# Environmentally-induced epigenetic conversion of a piRNA cluster

Karine Casier[1], Valérie Delmarre[1], Nathalie Gueguen[2], Catherine Hermant[1†], Elise Viodé[1], Chantal Vaury[2], Stéphane Ronsseray[1], Emilie Brasset[2], Laure Teysset[1]*, Antoine Boivin[1]*

[1]Laboratoire Biologie du Développement, UMR7622, Sorbonne Université, CNRS, Institut de Biologie Paris-Seine, Paris, France; [2]GReD, Université Clermont Auvergne, CNRS, INSERM, BP 10448, Clermont-Ferrand, France

**Abstract** Transposable element (TE) activity is repressed in animal gonads by PIWI-interacting RNAs (piRNAs) produced by piRNA clusters. Current models in flies propose that germinal piRNA clusters are functionally defined by the maternal inheritance of piRNAs produced during the previous generation. Taking advantage of an inactive, but ready to go, cluster of *P*-element derived transgene insertions in *Drosophila melanogaster*, we show here that raising flies at high temperature (29°C) instead of 25°C triggers the stable conversion of this locus from inactive into actively producing functional piRNAs. The increase of antisense transcripts from the cluster at 29°C combined with the requirement of transcription of euchromatic homologous sequences, suggests a role of double stranded RNA in the production of *de novo* piRNAs. This report describes the first case of the establishment of an active piRNA cluster by environmental changes in the absence of maternal inheritance of homologous piRNAs.

**Editorial note:** This article has been through an editorial process in which the authors decide how to respond to the issues raised during peer review. The Reviewing Editor's assessment is that all the issues have been addressed (see decision letter).

DOI: https://doi.org/10.7554/eLife.39842.001

*For correspondence:
laure.teysset@upmc.fr (LT);
antoine.boivin@upmc.fr (AB)

Present address: †Center for Interdisciplinary Research in Biology (CIRB), Collège de France, CNRS, INSERM, PSL Research University, Paris, France

Competing interests: The authors declare that no competing interests exist.

## Introduction

Transposable element (TE) activity needs to be repressed to avoid severe genome instability and gametogenesis defects. In humans, growing evidence has implicated TE in several disorders such as cancers defining a new field of diseases called transposopathies (*Wylie et al., 2016a*; *Wylie et al., 2016b*). In the animal germline, TE activity is controlled at both transcriptional and post-transcriptional levels by small RNAs called piRNAs associated with the PIWI clade of germline Argonaute proteins (Piwi, Aub and Ago3 in *Drosophila*) (*Brennecke et al., 2007*; *Gunawardane et al., 2007*; *Le Thomas et al., 2013*; *Sienski et al., 2012*). piRNAs are processed from transcripts produced from specific heterochromatic loci enriched in TE fragments, called piRNA clusters (*Brennecke et al., 2007*; *Gunawardane et al., 2007*). These loci undergo non-canonical transcription, ignoring splicing and transcription termination signals, licensed by specific protein complexes such as Rhino-Deadlock-Cutoff (*Mohn et al., 2014*; *Zhang et al., 2014*) and Moonshiner-TRF2 (*Andersen et al., 2017*). Thus, when a new TE inserts into a naive genome, it will freely transpose until one copy gets inserted into a piRNA cluster leading to the production of homologous new TE piRNAs that will then repress transposition (*Brennecke et al., 2008*). In support of this idea, exogenous sequences inserted into preexisting piRNA clusters lead to the production of matching piRNAs (*de Vanssay et al., 2012*; *Hermant et al., 2015*; *Marie et al., 2017*; *Muerdter et al., 2012*; *Pöyhönen et al., 2012*). The specificity of the efficient repression mediated by piRNAs appears to be determined solely by the piRNA cluster sequences. Thus, it raises the question of how piRNA

cluster loci are themselves specified. Histone H3 lysine nine tri-methylation (H3K9me3) that is recognized by Rhino, a paralog of heterochromatin protein HP1 (*Klattenhoff et al., 2009*), is a shared feature of piRNA clusters. Enrichment of H3K9me3, however, is not specific to piRNA clusters and tethering Rhino onto a transgene leads to the production of piRNAs only when both sense and antisense transcripts are produced (*Zhang et al., 2014*). This suggests that neither H3K9me3 marks nor having Rhino-bound is sufficient to induce piRNA production. One current model proposes that piRNAs clusters are defined and activated at each generation by the deposition in the egg of their corresponding piRNAs from the mother (*Huang et al., 2017*). In support of this model, we previously described the first case of a stable transgenerational epigenetic conversion known as paramutation in animals (*de Vanssay et al., 2012*). This phenomenon was first described in plants and defined as "*an epigenetic interaction between two alleles of a locus, through which one allele induces a heritable modification of the other allele without modifying the DNA sequence*" (*Brink, 1956*; *Chandler, 2007*). In our previous study, we showed that an inactive non-producing piRNA cluster of *P* transgene insertions inherited from the father can be converted into a piRNA-producing cluster by piRNAs inherited from the mother (*de Vanssay et al., 2012*). This attractive model, however, does not answer the question of how the first piRNAs were produced.

To address this paradox, we used the same *BX2* cluster of seven *P(lacW)* transgenes, which resulted from multiple and successive *P(lacW)* transposition events, thus resembling the structure of natural piRNA clusters (*Dorer and Henikoff, 1994*). The key advantage of the *BX2* locus is that it can exist in two epigenetic states for the production of germline piRNAs: 1) the inactive state ($BX2^{OFF}$) does not produce any piRNAs and thus is unable to repress the expression of homologous sequences, and 2) the active state ($BX2^{ON}$) produces abundant piRNAs that functionally repress a homologous reporter transgene in the female germline (*de Vanssay et al., 2012*; *Hermant et al., 2015*). We therefore used *BX2* in an inactive state to search for conditions that would convert it into an active piRNA-producing locus, without pre-existing maternal piRNAs. In this report, we describe how culturing flies at high temperature, 29°C instead of 25°C, induces the conversion of an inactive *BX2* locus ($BX2^{OFF}$) into a stable piRNA cluster exhibiting repression properties ($BX2^{ON}$). It should be noted that flies in their natural habitat exhibit this range of temperature, especially in the context of global warming. These data provide the first report of a *de novo* piRNA cluster establishment independent of maternal inheritance of homologous piRNAs and highlight how environmental changes can stably induce transgenerational modification of the epigenome.

## Results

### Germline silencing induced at high temperature

Earlier studies of hybrid dysgenesis reported that high temperature enhances *P*-element repression (*Ronsseray et al., 1984*) and that thermic modification of *P* repression can persist over several generations (*Ronsseray, 1986*). Moreover, *P*-element repression in a strain carrying two *P*-elements inserted into a subtelomeric piRNA cluster can be stimulated by heat treatment (*Ronsseray et al., 1991*). Very recently, the tracking of natural invasion of *P* elements in *Drosophila simulans* confirmed the key role of high temperature in the establishment of repression through generations (*Kofler et al., 2018*). These results suggested that temperature may influence the activity of some piRNA clusters. To investigate whether high temperature (29°C) could affect the stability of *BX2* epialleles ($BX2^{OFF}$ and $BX2^{ON}$) across generations, we generated flies carrying, on the same chromosome, each of the *BX2* epialleles and an euchromatic reporter transgene sharing *P* and *lacZ* sequences with *BX2* (made of seven *P(lacW)*, *Figure 1—figure supplement 1A*). This transgene promotes the expression of ß-Galactosidase both in the germline and in the somatic cells of the ovary and thus will hereinafter be referred to as '*P(TARGET)^{GS}*' (*Figure 1—figure supplement 1B*). As was previously described (*de Vanssay et al., 2012*), at 25°C $BX2^{OFF}$ does not synthesize functional piRNAs complementary to *P(TARGET)^{GS}* resulting in ß-Galactosidase expression in whole ovaries of $BX2^{OFF}$, *P(TARGET)^{GS}* lines (*Figure 1A*). Whereas in $BX2^{ON}$, *P(TARGET)^{GS}* lines, functional *lacZ* piRNAs are synthesized in the germline where they specifically repress the *P(TARGET)^{GS}* ß-Galactosidase expression (*Figure 1B*). Both $BX2^{OFF}$, *P(TARGET)^{GS}* and $BX2^{ON}$, *P(TARGET)^{GS}* lines incubated at 25°C for 23 generations maintained their epigenetic state, showing that both epialleles are stable (*Figure 1D*, *Supplementary file 1*). At 29°C, the repression capacity of $BX2^{ON}$, *P(TARGET)^{GS}* lines

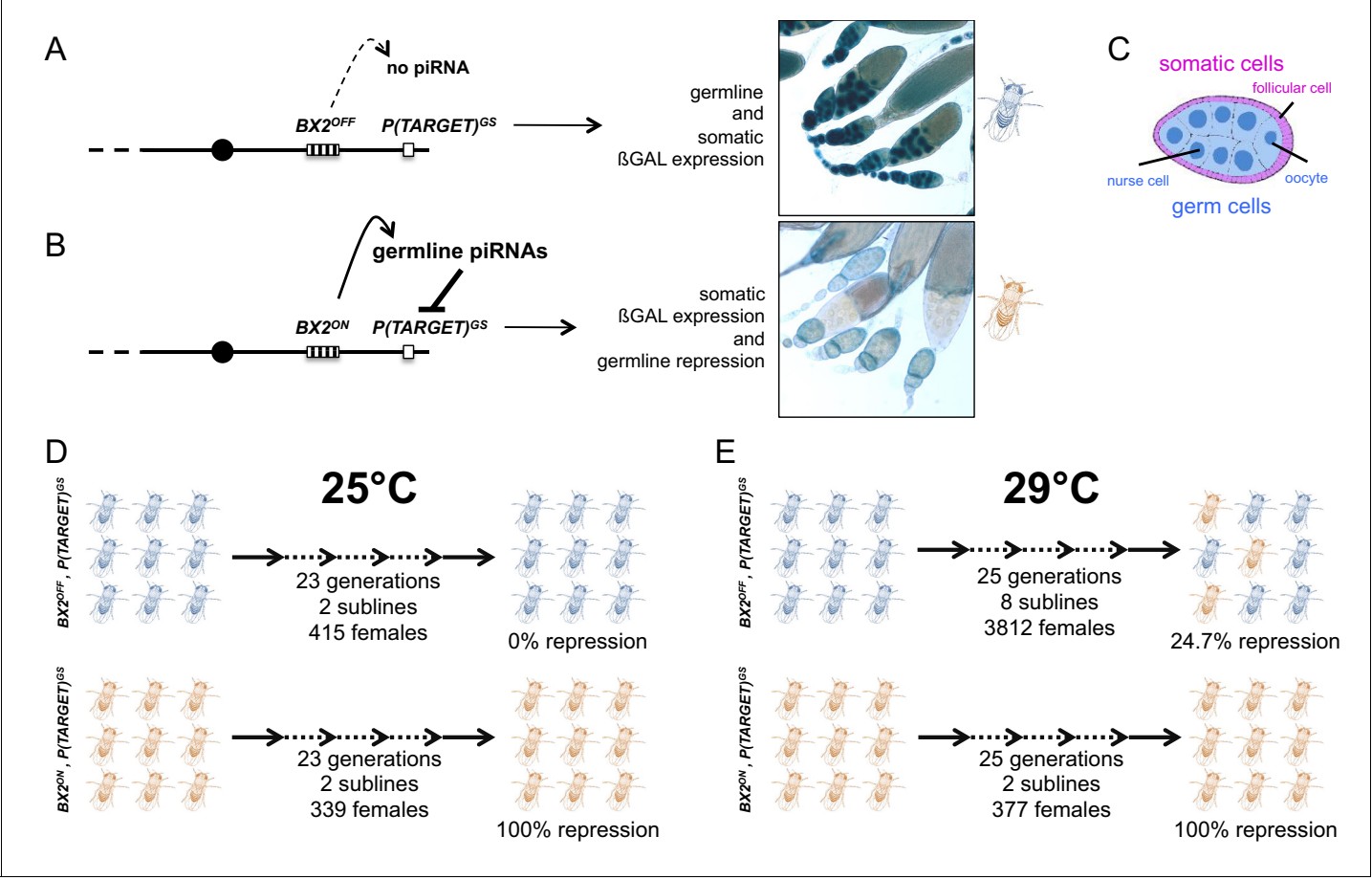

**Figure 1.** Functional assay of the *BX2* epigenetic state. Females carrying either one of the *BX2* epialleles and *P(TARGET)*$^{GS}$ were analyzed. (A) When *BX2* is OFF for production of piRNAs (*BX2*$^{OFF}$), no repression of *P(TARGET)*$^{GS}$ occurs, allowing expression of ß-Galactosidase in both germline and in somatic lineages in ovaries (ß-Galactosidase staining). *BX2*$^{OFF}$ is illustrated by a blue fly. (B) When *BX2* is ON for production of piRNAs (*BX2*$^{ON}$), repression of *P(TARGET)*$^{GS}$ occurs only in the germline lineage. *BX2*$^{ON}$ is illustrated by a light brown fly. (C) Drawing of an intermediate egg chamber showing germ cells (nurse cells and oocyte in blue) surrounded by somatic follicular cells (in pink), adapted from *Figure 1A* from *Frydman and Spradling (2001)*. (D) At 25°C, *BX2*$^{OFF}$ and *BX2*$^{ON}$ are stable over generations. (E) At 29°C, *BX2*$^{OFF}$ can be converted into *BX2*$^{ON}$, while *BX2*$^{ON}$ is stable over generations.

DOI: https://doi.org/10.7554/eLife.39842.002

The following figure supplements are available for figure 1:

**Figure supplement 1.** Schematic maps, drawn at scale, of the different transgenes used in this study.
DOI: https://doi.org/10.7554/eLife.39842.003

**Figure supplement 2.** Maternal (A) and paramutagenic (B) effect of *BX2*$^{ON}$ lines.
DOI: https://doi.org/10.7554/eLife.39842.004

**Figure supplement 3.** Dynamic of conversion across generations at 29°C.
DOI: https://doi.org/10.7554/eLife.39842.005

remained stable through 25 generations. Among the *BX2*$^{OFF}$, *P(TARGET)*$^{GS}$ lines, 24.7% of females analyzed during 25 generations showed a complete and specific germline ß-Galactosidase repression (n = 3812, *Figure 1E* and *Supplementary file 1*), suggesting a conversion of the *BX2*$^{OFF}$ epiallele into *BX2*$^{ON}$. Interestingly, the appearance of females showing ß-Galactosidase repression was gradual and stochastic, resulting in a global frequency that increased with the number of generations (*Supplementary file 1*). To test whether the temperature-induced conversion was stable, a set of five lines showing full repression capacity after 23 generations at 29°C, obtained from an independent experiment, were transferred to 25°C and tested for their silencing capacities for several generations. In all cases, the silencing capacities of the *BX2*$^{ON}$ epiallele induced at 29°C remained stable

during 50 additional generations at 25°C (*Supplementary file 2*). These stable $BX2^{ON}$ lines converted by high temperature were named hereafter $BX2^{\Theta}$ (Greek theta for temperature) to distinguish them from the $BX2^*$ lines converted by maternally inherited piRNAs (*de Vanssay et al., 2012*). Taken together, our data show that $BX2^{OFF}$ can be functionally converted by high temperature (29°C), strongly suggesting that *de novo* piRNA production can occur in the absence of maternal inheritance of homologous piRNAs.

## *BX2* lines converted by high temperature or by maternal homologous piRNA inheritance present identical functional and molecular properties

We further characterized the functional and molecular properties of $BX2^{\Theta}$ activated by temperature and compared them to $BX2^*$ activated by maternal inheritance of homologous piRNAs. Firstly, we compared the maternal and paternal *BX2* locus inheritance effect of three $BX2^{\Theta}$ lines and three $BX2^*$ at 25°C (*Figure 1—figure supplement 2A*). Maternal inheritance of either $BX2^{\Theta}$ or $BX2^*$ loci leads to complete and stable repression of ß-Galactosidase expression (n flies = 152 and 159, respectively, *Supplementary file 3*), whereas paternal inheritance of either $BX2^{\Theta}$ or $BX2^*$ loci, that is in absence of maternal piRNA deposition, results in ß-Galactosidase expression, and thus a definitive loss of *BX2* silencing capacities (n flies = 156 and 155, respectively, *Supplementary file 3*). Secondly, we previously showed that progeny with a paternally inherited $BX2^{OFF}$ locus and maternally inherited $BX2^*$ piRNAs, but lacking the maternally $BX2^*$ genomic locus, have 100% conversion (*de Vanssay et al., 2012*). This process of recurrent conversions of an allele that is heritable without DNA modification is known as paramutation, thus $BX2^*$ females are paramutagenic, that is able to trigger paramutation. To test this property on *BX2* lines converted by temperature, $BX2^{\Theta}$ females were crossed with $BX2^{OFF}$ males. The progeny that inherited the paternal $BX2^{OFF}$ locus but not the maternal $BX2^{\Theta}$ locus were selected and three independent lines were established (*Figure 1—figure supplement 2B*). Silencing measured over 20 generations revealed 100% of repression capacity showing that $BX2^{\Theta}$ is also paramutagenic (n flies = 159, *Supplementary file 4*).

To determine whether the silencing capacities of $BX2^{\Theta}$ involved piRNAs, small RNAs from $BX2^{OFF}$, $BX2^{\Theta}$ and $BX2^*$ ovaries were extracted and sequenced (*Supplementary file 5*). Unique reads matching the *P(lacW)* sequences were identified only in the $BX2^{\Theta}$ and $BX2^*$ libraries (*Figure 2A*). Most of these small RNAs display all of the characteristics of *bona fide* germline piRNAs, that is a high proportion of 23–29 nt with a strong U bias on the first 5′ nucleotide, an enrichment of a 10 nucleotide overlap between sense and antisense piRNAs, also known as the ping-pong signature, and a high proportion of reads with A at the tenth position among the 10 nt overlapped reads (*Figure 2B–C*) (*Brennecke et al., 2007*; *Gunawardane et al., 2007*). As a control, the *42AB* piRNA cluster, a canonical germline dual-strand piRNA cluster, presented no significant difference between the three genotypes (*Figure 2—figure supplement 1*). Therefore, these results show that high temperature can initiate piRNA production from *BX2* naive sequences ($BX2^{OFF}$) and strongly suggest that once a piRNA cluster is activated for piRNA production, the 'ON' state is maintained at each generation by maternal inheritance of piRNAs.

*BX2* is inserted into the first intron of the *AGO1* gene (*de Vanssay et al., 2012*) and we looked at the piRNA production from this region in the different *BX2* epigenetic contexts. No significant amount of piRNAs coming from the *AGO1* gene region can be detected whatever the *BX2* state (*Figure 2—figure supplement 2*). These findings indicate that the *AGO1* gene region is not a natural piRNA cluster. To test whether other non-piRNA producing genomic loci have started to produce piRNAs following high temperature treatment, we looked for specific piRNAs (23–29 nt) matching at unique positions on *Drosophila* chromosomes and compared them between $BX2^{\Theta}$ and $BX2^{OFF}$. The reads were then resampled per 50 kilobases windows. To eliminate background noise, only regions that produced more than five piRNAs per kilobase on average in both libraries were considered. Only exons of the *white* gene present in the *P(lacW)* transgenes of *BX2* showed differential piRNA expression (log2 ratio >8.5, *Figure 2—figure supplement 3*). This analysis revealed that the activation of piRNA production after thermic treatment is restricted to the *BX2* locus, suggesting that all other loci able to produce piRNAs are already active.

Previous studies had suggested that the chromatin state plays a role in the differential activity of *BX2* (*Le Thomas et al., 2014*). We therefore profiled H3K9me3 marks and Rhino binding on the *P(lacW)* transgene in ovaries from $BX2^{OFF}$, $BX2^{\Theta}$ and $BX2^*$ strains by chromatin immunoprecipitation (ChIP) followed by quantitative PCR (qPCR). In both strains $BX2^{\Theta}$ and $BX2^*$, H3K9me3 and Rhino

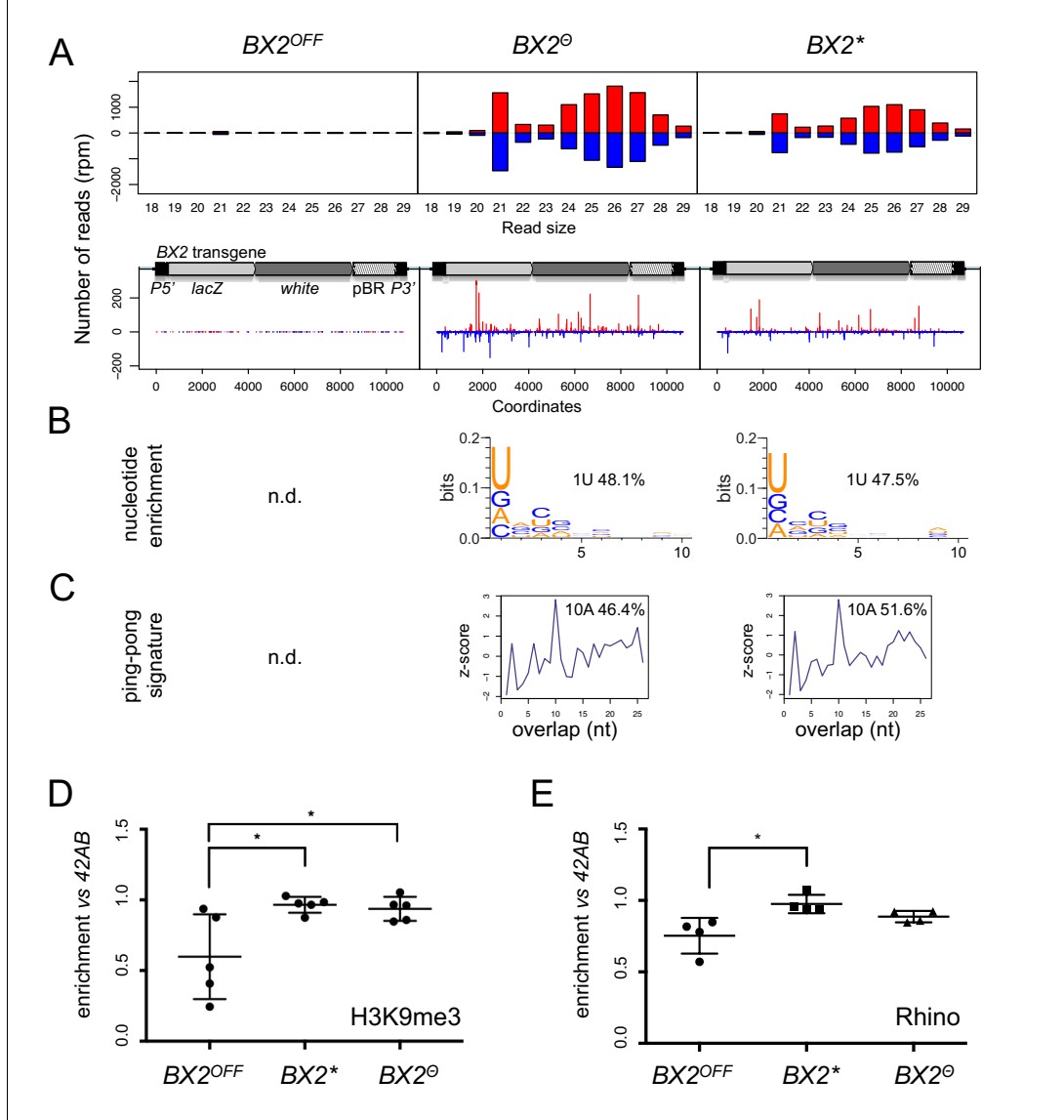

**Figure 2.** *BX2*⊖ and *BX2** produce piRNAs and are enriched in H3K9me3. (**A**) Size distribution of ovarian small RNAs matching *BX2* transgene sequences reveals that both *BX2*⊖ and *BX2** but not *BX2*^OFF produce 21 nt siRNAs and 23–29-nt piRNAs (upper panels, pBR = backbone plasmid pBR322). Positive and negative values correspond to sense (red) and antisense (blue) reads, respectively. Unique 23–29 nt mappers are shown on the *BX2* transgene sequences (lower panels). (**B**) Percentage of 23–29 nt small RNAs from *BX2*⊖ and *BX2** matching transgene sequence with a U at the first position are shown. n.d.: not determined due to low number of reads. (**C**) Relative frequency (z-score) of overlapping sense-antisense 23–29 nt RNA pairs reveals an enrichment of 10 nt overlapping corresponding to the ping-pong signature. (**D**) H3K9me3 and (**E**) Rhino binding on the *BX2* transgene in ovaries of *BX2*^OFF, *BX2*⊖ and *BX2** strains revealed by chromatin immunoprecipitation (ChIP) followed by quantitative PCR (qPCR) on specific *white* sequences. In both 'ON' strains, *BX2*⊖ and *BX2**, H3K9me3 and Rhino levels over the transgene are very similar and higher than in the *BX2*^OFF strain (unpaired *t*-test was used to calculate significance of the differences ($p < 0.05$, n = 5).

DOI: https://doi.org/10.7554/eLife.39842.006

The following figure supplements are available for figure 2:

**Figure supplement 1.** *42AB* piRNA production is not altered in different *BX2* epigenetic contexts.
DOI: https://doi.org/10.7554/eLife.39842.007

**Figure supplement 2.** The *AGO1* gene region does not produce piRNA by itself.
DOI: https://doi.org/10.7554/eLife.39842.008

**Figure supplement 3.** Mapping the genome for regions stably producing new piRNAs.
DOI: https://doi.org/10.7554/eLife.39842.009

were similarly enriched over the *P(lacW)* transgene compared to the *BX2^OFF* strain, significantly for H3K9me3 (*Figure 2D–E*). Taken together, these results show that *de novo* activation of *BX2^OFF* by 29°C treatment (*BX2^Θ*) or paramutation by maternal inheritance of homologous piRNAs (*BX2\**) lines leads to similar functional and molecular properties.

## Epigenetic conversion at 29°C occurs at a low rate from the first generation

To explain the low occurrence and the generational delay of *BX2* conversion at 29°C (*Supplementary file 1*), we propose that conversion is a complete but rare event occurring in a small number of egg chambers at each generation. Under this hypothesis, the sampling size of tested females should be crucial to observe such stochastic events. We therefore increased the number of analyzed females raised at 29°C during one generation. For this, eggs laid by females maintained at 25°C carrying the *P(TARGET)^GS* reporter transgene and the *BX2^OFF* locus were collected during three days. These eggs were then transferred at 29°C until adults emerged (*Figure 3A*). To follow their off-spring, we individually crossed 181 G1 females with two siblings and let them lay eggs for three days at 25°C. These 181 G1 females were then stained for ß-Galactosidase expression. Strikingly, repression occurred only after one generation at 29°C in a few of the egg chambers of 130 G1 females ($\approx$ 2.7% of the estimated total number of G1 egg chambers n $\approx$ 21700, *Figure 3B*, right panel). These results support our hypothesis whereby epigenetic conversion of *BX2^OFF* into *BX2^Θ* is an instantaneous and complete event occurring at a low frequency per egg chamber and at each generation that is kept at 29°C.

To test the stability of the epigenetic *BX2^Θ* states observed in G1 females, offspring daughters (G2) were raised at 25°C and their ovaries examined for ß-Galactosidase expression. Among G2 females, partial (n = 24) or complete (n = 17) repression of ß-Galactosidase expression in the germline was observed only in the progeny of those 130 G1 females in which partial repression was previously detected (*Figure 3C*). The proportion of 2.2% of converted G2 females (41/1830) is reminiscent with the proportion of repressed egg chambers observed in G1. The progeny of the 51 G1 females that did not present repression (*Figure 3B* left panel) did not show spontaneous conversion (*Figure 3C*). Taken together, these observations strongly suggest that newly converted *BX2^ON* egg chambers give rise to adult females with complete or partial silencing capacities. The low conversion rate observed in thousands of flies after one generation raised at high temperature and its stability through the next generation might explain the apparent delay of *BX2^ON* conversion of dozens of flies continuously raised at 29°C observed in the first set of experiments (see *Figure 1E* and *Supplementary file 1*). We more finely analyzed the silencing capacities of eight independent *BX2^OFF* lines throughout generations at 29°C (*Supplementary file 1*) by monitoring *P(TARGET)^GS* repression in each egg chamber instead of whole ovaries (*Supplementary file 6*). *BX2* conversion occurred in each tested line with various dynamics (*Figure 1—figure supplement 3A*), likely reflecting a genetic drift due to the low conversion occurrence coupled to an important sampling effect at each generation. Globally, the mean of repression frequencies seems to indicate a progressive increase of *BX2* conversion over generations (*Figure 1—figure supplement 3B*). Altogether, our results illustrate how environmental modifications like high temperature experienced during one generation might stably modify the epigenome of the future ones. Such a newly acquired epigenetic state may spread in a given population within a few generations.

## High temperature increases *BX2* antisense RNA but not piRNAs

Previously, we showed that *BX2^OFF* and *BX2^ON* produce similar amounts of sense and antisense transcripts (*de Vanssay et al., 2012*). However, these transcripts do not lead to ß-Galactosidase expression in the germline nor piRNA production in the *BX2^OFF* line. We wondered if high temperature might change the RNA steady-state level of *BX2*. To ensure that we were detecting RNA specifically from *BX2*, qRT-PCR experiments targeting the *lacZ* gene were carried out on ovarian RNA extracted from the *BX2^OFF* line that did not contain *P(TARGET)^GS*. We observed a significant increase in the steady-state *BX2* RNA levels at 29°C compared to 25°C (*Figure 4A*). Remarkably, strand-specific qRT-PCR experiments revealed that only *BX2* antisense transcripts, corresponding to antisense *lacZ* transcripts, increase at 29°C (*Figure 4B*). As *BX2* is inserted into the *AGO1* gene in a convergent transcription manner (*de Vanssay et al., 2012*, *Figure 4C*), we compared *AGO1* steady-state RNA

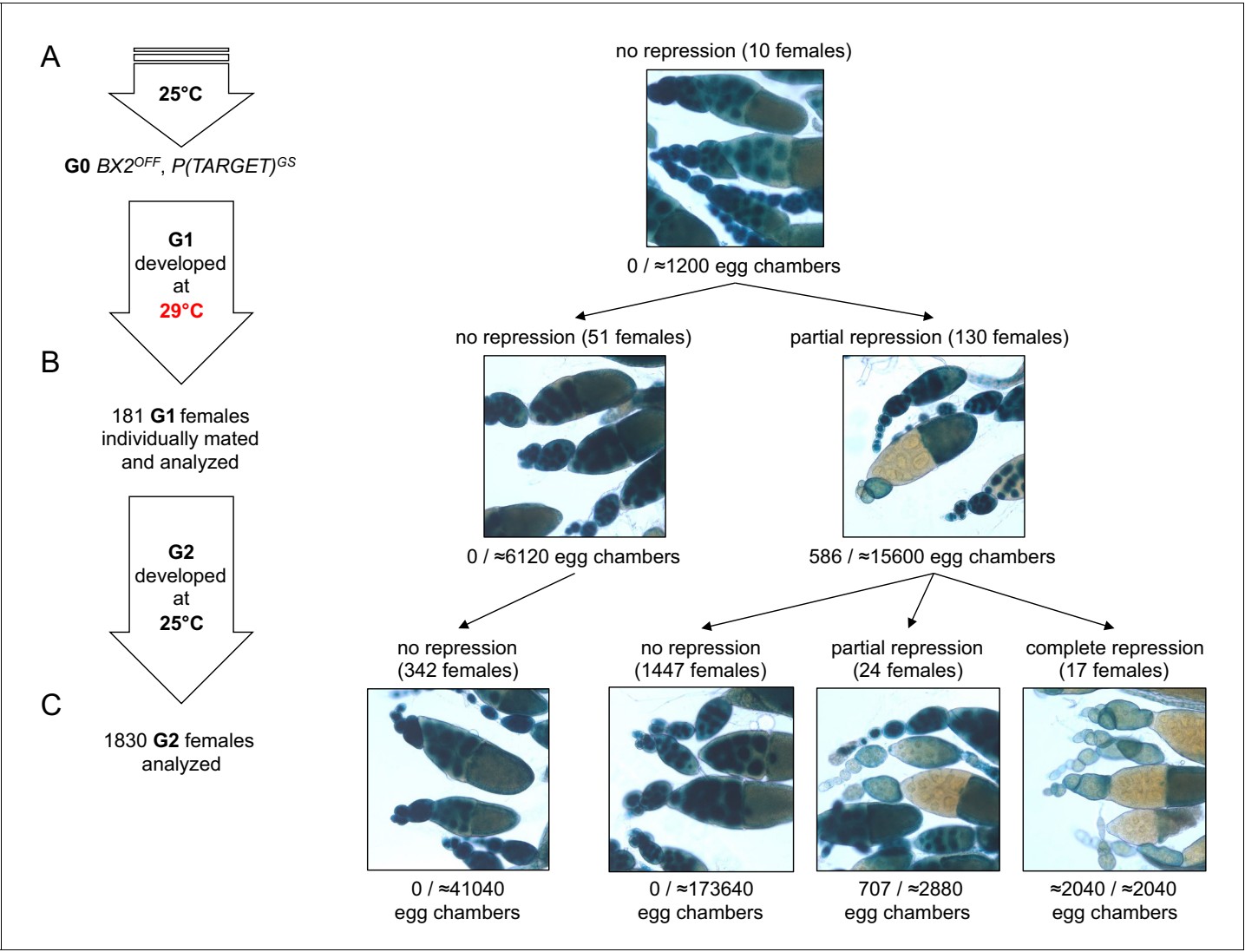

**Figure 3.** *BX2* conversion at 29°C occurs in one generation at a low rate. (**A**) G0 females carrying the *P(TARGET)*$^{GS}$ reporter and *BX2*$^{OFF}$ laid eggs at 25°C during three days. The *BX2*$^{OFF}$ state of these females was confirmed after the three days at 25°C by ß-Galactosidase staining (number of egg chambers ≥ 1200). (**B**) Their eggs were allowed to develop at 29°C until emergence of the next generation. G1 females (n = 181) were individually mated with two siblings and left to lay for three days at 25°C. G1 females were then individually stained for ß-Galactosidase expression. Strikingly, 130 females (71.8%) show ß-Galactosidase repression in some egg chambers (586 among ≈21700 - estimation of the total egg chamber number among 181 females). The *BX2*$^{OFF}$ into *BX2*$^{ON}$ conversion frequency is ≈2.7%. (**C**) Analysis of each G1 female progeny developed at 25°C by ß-Galactosidase staining. The progeny of the 51 G1 females that did not present repression maintained *BX2*$^{OFF}$ state (n flies = 342). Most of the progeny of the 130 G1 females presenting conversion show no repression (97.2%, n flies = 1488) while 41 females present partial (n = 24) or complete (n = 17) repression of the germline expression of ß-Galactosidase.

DOI: https://doi.org/10.7554/eLife.39842.010

The following figure supplements are available for figure 3:

**Figure supplement 1.** No *BX2* conversion at 29°C without the *P(TARGET)*$^{GS}$ reporter transgene.

DOI: https://doi.org/10.7554/eLife.39842.011

**Figure supplement 2.** No *BX2* conversion at 29°C with a transgene that is not expressed in the germline.

DOI: https://doi.org/10.7554/eLife.39842.012

**Figure supplement 3.** *BX2*$^{ON}$ silencing depends on *moonshiner*.

DOI: https://doi.org/10.7554/eLife.39842.013

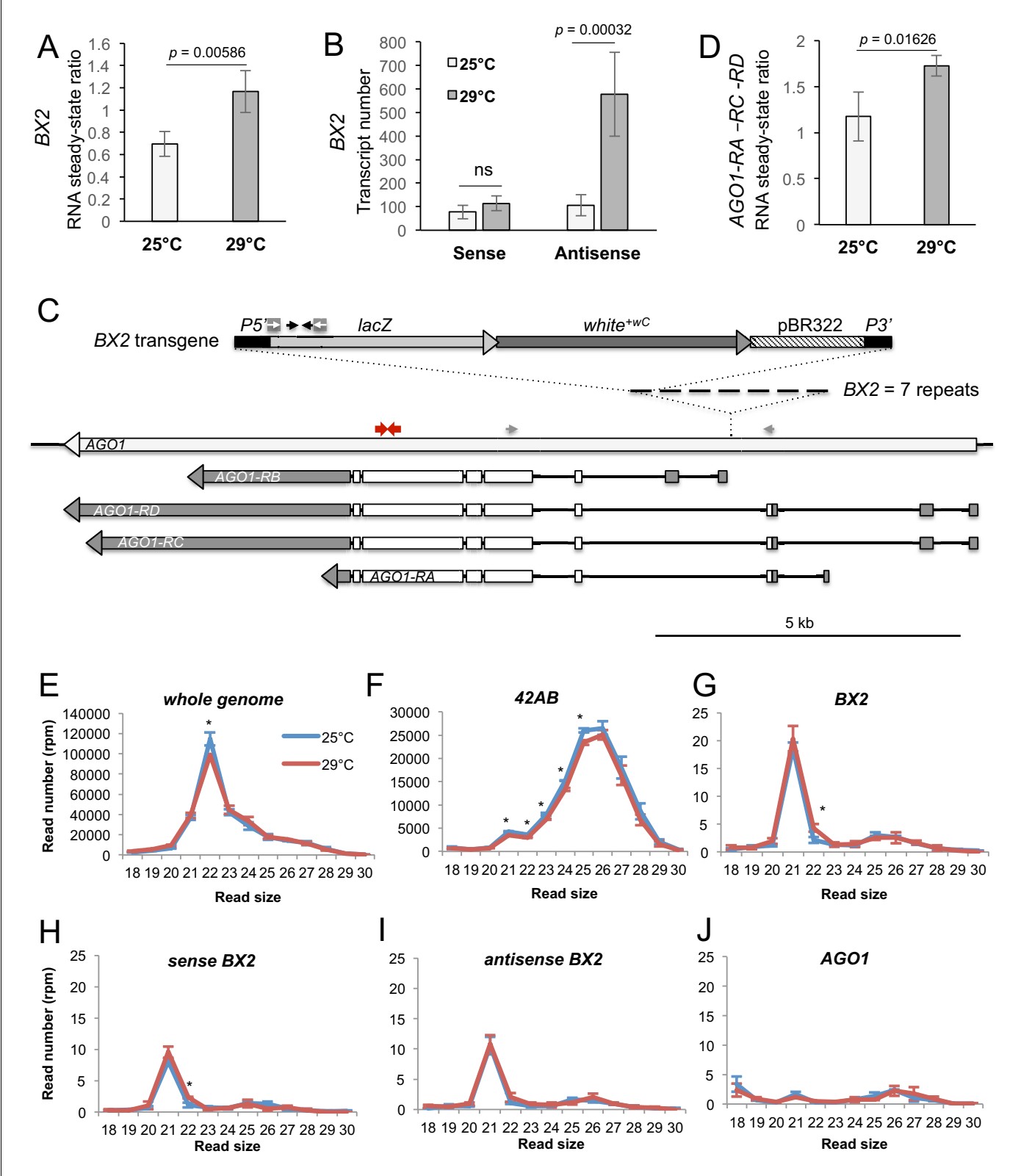

**Figure 4.** $BX2^{OFF}$ antisense RNA increase at 29°C. (**A**) RT-qPCR experiments revealed that the steady-state level of ovarian *lacZ* RNAs from *BX2* is more abundant at 29°C (n = 5) compared to 25°C (n = 6). (**B**) Sense-specific RT-qPCR experiments revealed that only antisense transcripts from *BX2*, corresponding to antisense *lacZ* transcripts, are increased (25°C n = 6, 29°C n = 4). Significant *p*-values are given (bilateral Student's *t*-test). ns: not significant. (**C**) Map of the *BX2* locus containing seven *P(lacW)* transgenes inserted into the *AGO1* gene. *P(lacW)* and *AGO1* are drawn to scale. The *lacZ*

*Figure 4 continued on next page*

*Figure 4 continued*

gene contained in *P(lacW)* and *AGO1* are transcriptionally in opposite directions. Black arrows show *lacZ* primers used for (**A**) and (**B**) experiments. White arrows show primers used for sense-specific reverse transcription in (**B**) experiment. Grey arrows show *AGO1* primers used for (**D**) experiment: these primers are specific for *AGO1* transcripts (RA, RC and RD) that originate from promoters located upstream the *BX2* insertion point and, thus, are potentially convergent to *BX2*. Red arrows show primers used to measure steady-state of all *AGO1* isoforms (see *Figure 4—figure supplement 1D*). (**D**) RT-qPCR experiments performed on flies devoid of *P(lacW)* transgenes (*w1118* context) revealed that the steady-state level of ovarian *AGO1-RA, -RC* and *-RD* RNA isoforms is more abundant at 29°C (n = 5) compared to 25°C (n = 6). (**E–I**) To compare small RNAs at 25 *versus* 29°C, total RNAs were extracted from *BX2OFF* ovaries dissected from adults incubated at 25°C or 29°C. Three samples were tested for each temperature. Small RNAs from 18 to 30 nt were deep sequenced. For each library, normalization has been performed for 1 million reads matching the *Drosophila* genome (rpm, *Supplementary file 7*). Size distributions of unique reads that match reference sequences are given. (**E**) Small RNAs matching the *Drosophila* genome present similar profiles in both temperatures except for 22 nt RNAs that are more represented at 25°C. (**F**) The 21 to 25 nt reads matching the *42AB* piRNA cluster that range from 21 to 25 nt are slightly more abundant at 25°C. (**G**) Strikingly, almost only 21 nt RNAs match *BX2* sequence. They are equally distributed among sense (**H**) and antisense (**I**) sequences at both temperatures. (**J**) No small RNAs corresponding to the *AGO1* gene can be detected whatever the temperature. *=p < 0.05, bilateral Student's *t*-test.

DOI: https://doi.org/10.7554/eLife.39842.014

The following figure supplement is available for figure 4:

**Figure supplement 1.** Complementary RNA steady-state measurements by quantitative RT-PCR.

DOI: https://doi.org/10.7554/eLife.39842.015

level at 29°C and 25°C. *AGO1* transcript isoforms that are initiated upstream the *BX2* insertion point are significantly increased at 29°C (*Figure 4D* and *Figure 4—figure supplement 1A–B*). Thus, it is possible that, at 29°C, an increase of transcription from the *AGO1* promoters located upstream the *BX2* insertion point could lead to an increase of *BX2* antisense RNA transcription.

We then examined whether the increase of *BX2* antisense RNAs leads to an increase of antisense small RNAs. Ovarian small RNAs (18 to 30 nucleotides) of *BX2OFF* flies (without *P(TARGET)GS*) raised at 25°C and at 29°C for one generation were sequenced and the read numbers normalized (*Supplementary file 7*). A slight, yet statistically significant, decrease is observed at 29°C for small RNAs matching the whole genome (*Figure 4E*) and for the *42AB* piRNA cluster (*Figure 4F*). Strikingly, no piRNAs were produced from the *BX2* locus at 25°C nor after one generation at 29°C. Thus, the increase of *BX2* antisense transcripts observed at 29°C (*Figure 4B*) did not correlate with an increase of corresponding antisense piRNAs. At 25°C and 29°C, *BX2OFF* produced the same low amount of 21 nt small RNAs, equally distributed between sense and antisense (*Figure 4G,H,I*), suggesting that *BX2* transcripts are processed into siRNAs. These results confirm that *AGO1* is not a piRNA producing locus (as shown in *Figure 2—figure supplement 2*) and showed that, at 29°C, *AGO1* is still not producing small RNAs (*Figure 4J*). These data indicate that 21 nt small RNA production was restricted to *BX2* sequences.

## Heat conversion requires a transcribed homologous sequence in trans

Quantitative RT-PCR experiments described above were carried out on flies bearing only the *BX2* locus while all conversion experiments at 29°C were done with flies bearing the *BX2* and the *P(TARGET)GS* locus. Interestingly, the amount of *P(TARGET)GS* transcripts is affected by temperature but in the opposite way to *BX2*, as less transcripts were measured at 29°C compared to 25°C (*Figure 4—figure supplement 1C*). We next asked whether the *P(TARGET)GS* transgene could participate in the conversion process of *BX2*. For this, 'heat-activated-conversion' experiments of *BX2* were done in flies not carrying the *P(TARGET)GS* (*Figure 3—figure supplement 1*). To assess the *BX2* epigenetic state of the G1 raised at 29°C, 157 G1 females were individually crossed at 25°C with males harboring the *P(TARGET)GS* transgene. Among the 1137 G2 females analyzed, only one female presented partial repression of the ß-Galactosidase expression and none presented complete repression (*Figure 3—figure supplement 1*). If we compare these results with those obtained with the *BX2, P(TARGET)GS* lines (*Figure 3*), the difference was highly significant (p=8.5×10⁻⁶, homogeneity χ² = 23.35 with 2 degrees of freedom, *Supplementary file 8*). To further validate the requirement of the euchromatic homologous transgene *P(TARGET)GS* in establishing the temperature-dependent *BX2* conversion, we generated eight independent lines in which *BX2* was recombined into the same *P(TARGET)GS* genetic background but without the *P(TARGET)GS* transgene. After 30 generations at 29°C, no female showing ß-Galactosidase repression was observed (*Supplementary file 9*). A

homogeneity $\chi^2$ test comparing, at G13, the repression occurrence in $BX2$, $P(TARGET)^{GS}$ lines (31/161, **Supplementary file 1**) and in recombined $BX2$ lines (0/975, **Supplementary file 9**) is highly significant (p=7.04×10$^{-44}$, homogeneity $\chi^2$ = 192.9 with 1 degree of freedom), arguing against a background effect of the $P(TARGET)^{GS}$ line in the conversion phenomenon. Additionally, we reproduced the experiment with an euchromatic $P(TARGET)$ expressed only in the germline and referred to hereinafter as '$P(TARGET)^G$' (**Figure 1—figure supplement 1A–B**). In the presence of $P(TARGET)^G$, $BX2$ was converted at 29°C at a rate comparable with that observed with $P(TARGET)^{GS}$ (**Supplementary file 10**). These data show that $BX2$ conversion by temperature cannot be attributed to any specificity linked to the $P(TARGET)^{GS}$ insertion. To know if the transcription of the $P(TARGET)^{GS}$ (or of the $P(TARGET)^G$) is required, we reproduced the same experiment with another euchromatic transgene that is not expressed in the germline but in the somatic cells surrounding the germ cells and therefore referred to hereinafter as '$P(TARGET)^S$' (**Figure 1—figure supplement 1A–B**). In the presence of $P(TARGET)^S$, $BX2$ was not converted at 29°C (0/784, **Figure 3—figure supplement 2**). A homogeneity $\chi^2$ test comparing these results with those obtained with $P(TARGET)^{GS}$ considering only the complete $BX2^{ON}$ G2 females (17/1464 females, **Figure 3**) is significant (p=0.0055, homogeneity $\chi^2$ = 7.7 with 1 degree of freedom). We conclude that the transcription of a reporter transgene sharing homologous sequences (i.e. *lacZ*) with $BX2$ is required in the germline for $BX2$ conversion at 29°C.

To summarize, in the absence of $P(TARGET)$ sequences, at 29°C $BX2^{OFF}$ produces an elevated number of antisense transcripts, no piRNAs and is unable to be converted to $BX2^{ON}$. In contrast, when a $P(TARGET)$ is present and transcribed in the germline, $BX2$ conversion and piRNA production are observed at 29°C. Although much of the mechanistic aspects of $BX2$ conversion remain unknown, these findings lead us to propose that, at 29°C, double strand RNA (dsRNA) made of the excess of $BX2$ antisense transcripts and the sense $P(TARGET)$ transcripts could be a prerequisite for the production of *de novo* piRNAs and the conversion of $BX2$ into an active piRNA cluster (see recapitulative model **Figure 5**).

## Discussion

Here, we report on the heritable establishment of a new piRNA cluster associated with silencing properties induced by high temperature during development. The epigenetic response to heat exposure has been studied in several model species: in *Arabidopsis* for instance, increasing temperature induces transcriptional activation of repetitive elements (**Ito et al., 2011**; **Pecinka et al., 2010**; **Tittel-Elmer et al., 2010**). Whether these changes involve chromatin modifications is not clear but none of these modifications have been found to be heritable through generations except in mutants for siRNA biogenesis where high frequency of new TE insertions was observed in the progeny of stressed plants (**Ito et al., 2011**). In animals, response to heat can result in modification of DNA methylation at specific loci in reef building coral (**Dimond and Roberts, 2016**), chicken (**Yossifoff et al., 2008**) and wild guinea pigs (**Weyrich et al., 2016**). In the latter, modifications affecting ≈50 genes are inherited in G1 progeny (**Weyrich et al., 2016**). The mechanisms of this heritability, however, are not yet understood. In *Drosophila*, heat-shock treatment of 0–2 hr embryos for one hour at 37°C or subjecting flies to osmotic stress induce phosphorylation of dATF-2 and its release from heterochromatin (**Seong et al., 2011**). This defective chromatin state is maintained for several generations before returning to the original state. We have tested if such stresses were able to convert $BX2^{OFF}$ into $BX2^{ON}$ in one generation but neither heat-shock nor osmotic stress induces $BX2$ conversion (**Supplementary file 11**), suggesting that $BX2$ activation does not depend on dATF-2. In a more recent paper, Fast *et al.* (**Fast et al., 2017**) found less piRNAs at 29°C than at 18°C. However, RNAseq analyses of differentially expressed genes involved in the piRNA pathway were not conclusive, as some piRNA genes (*ago3*, *aub*, *zuc*, *armi*) were more expressed at 29°C while others (*shu*, *hsp83*, *Yb*) were less expressed as compared to the levels at 18°C (**Fast et al., 2017**). Overall, the enhancement of the piRNA ping-pong amplification loop observed at 29°C was attributed to RNA secondary structures, because of a lack of specificity for any particular class of TE (**Fast et al., 2017**). Furthermore, in contrast to our RT-qPCR results obtained at 25°C and 29°C (**Figure 4D** and **Figure 4—figure supplement 1A–B**), *AGO1* was not differentially expressed between 18°C and 29°C (**Fast et al., 2017**). This difference can be explained, because, in our experiments, we checked for specific spliced transcripts of *AGO1* (RA, RC and RD) that originate upstream

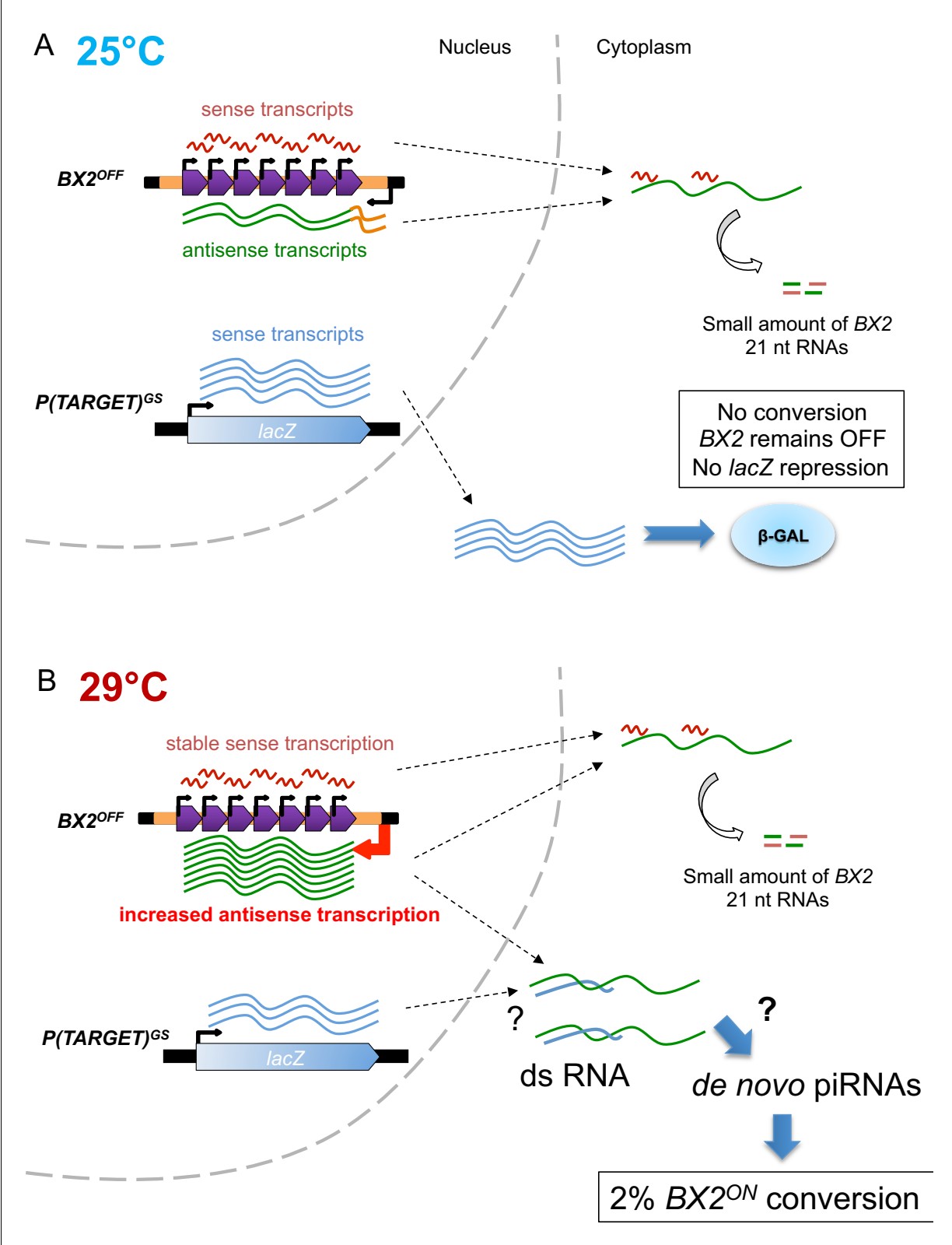

**Figure 5.** Model of *BX2* activation at 29°C. (**A**) At 25°C, a low bidirectional transcription of the *BX2* cluster leads to a small production of 21 nt RNAs. *BX2^OFF^* is stable at 25°C, no conversion event is observed and the *lacZ* reporter gene from *P(TARGET)^GS^* remains active throughout generations. (**B**) At 29°C, a specific increase in antisense transcription occurs, presumably due to a higher activity of the promoter of the *AGO1* gene (orange box). This excess of *BX2* antisense RNA could interact with sense *P(TARGET)^GS^* transcripts to produce double stranded RNA. Through a yet unknown mechanism,

*Figure 5 continued on next page*

Figure 5 continued

such dsRNA could lead to the formation of *de novo BX2* piRNAs. These piRNAs could in turn trigger the conversion of $BX2^{OFF}$ into an active piRNA cluster, a phenomenon observed based on the repression of the *lacZ* reporter gene of $P(TARGET)^{GS}$. The *BX2* conversion is a rare event ($\approx 2\%$ per generation) but once achieved, $BX2^{ON}$ remains active throughout generations due to the maternal inheritance of homologous piRNAs and the paramutation of the paternal $BX2^{OFF}$ allele.

DOI: https://doi.org/10.7554/eLife.39842.016

the *BX2* insertion point (**Figure 4C**). In agreement with Fast *et al.*, no statistical difference in the global amount of *AGO1* transcripts between 25°C and 29°C is observed using primers located downstream the *BX2* insertion point (**Figure 4—figure supplement 1D**). This suggests that the *AGO1* promoter located downstream the *BX2* insertion point, and responsible for the production of the RB isoform (see **Figure 4**), is not sensitive to temperature. Taken together, these observations show that temperature modification might induce epigenetic changes in several species but the underlying mechanisms remain largely unsolved.

Whole genome comparison of small RNA sequencing between $BX2^{OFF}$ and newly $BX2^{ON}$ heat-converted flies ($BX2^{\Theta}$) did not reveal additional regions stably converted for piRNA production (**Figure 2—figure supplement 3**). This suggests that no other loci are metastable for piRNA production, that is all potential piRNA clusters are already active at 25°C. This observation raises the question of what makes *BX2* locus competent for piRNA activation at high temperature. The *BX2* locus is the result of successive induced transpositions of *P(lacW)* used to screen for *white*-variegating phenotype (**Dorer and Henikoff, 1994**). Thus, *BX2* resembles natural TE clusters where TEs have the capacity to transpose into each other assembling structures named *nested TE*, as described in numerous genomes (**Gao et al., 2012**; **Liu et al., 2007**; **Zanni et al., 2013**). Some *nested TE* loci might have the capacity to be activated and respond to a new TE invasion. In *Drosophila*, *BX2*-like tandemly inserted transgenes were shown to be new sites of HP1 enrichment in larval salivary glands, emphasizing a heterochromatic structure at the *BX2* locus in somatic cells (**Fanti et al., 1998**), and to cause pairing-dependent silencing (**Dorer and Henikoff, 1997**). When tested for repressing capacities, however, this strain is inactive for *BX2* piRNA production (**de Vanssay et al., 2012**; **Josse et al., 2008**). We have shown that maternally inherited *P(lacW)* piRNAs are able to paramutate with complete and stable penetrance from an inactive *BX2* locus into an active locus for piRNA production (**de Vanssay et al., 2012**). The paramutated *BX2* locus appeared to be a genuine piRNA cluster since it is sensitive to a number of factors known to be involved in piRNA biogenesis such as *aub*, *rhi*, *cuff*, *zuc* (**Hermant et al., 2015**) and *moonshiner* (**Figure 3—figure supplement 3**). The number of transgene copies appears to be crucial in the process since smaller number of transgenes results in less somatic heterochromatinization (**Dorer and Henikoff, 1994**; **Fanti et al., 1998**), less pairing-dependent silencing (**Dorer and Henikoff, 1997**) and unstable paramutation (**de Vanssay et al., 2012**). Taken together, these data suggest that the heterochromatic structure of a cluster precedes piRNA production. This is supported by our ChIP experiments showing that H3K9me3 levels on $BX2^{OFF}$ are slightly below the H3K9me3 level of piRNA producing states ($BX2^{ON}$ and $BX2^{\Theta}$, **Figure 2D**). The same observation can be made for Rhino (**Figure 2E**), suggesting that Rhino may be already present on the $BX2^{OFF}$ locus but below the threshold required for piRNA production as suggested by **Akulenko et al. (2018)**. Thus, a locus made of repeated sequences and being likely heterochromatic (H3K9me3, Rhino) is a necessary but not sufficient condition to specify an active piRNA cluster.

In the germline, piRNA clusters produce piRNAs from both strands and it was recently shown that, in most cases, transcription initiates within clusters on both strands through the interaction of Rhino and Moonshiner (**Andersen et al., 2017**). In few cases, however, piRNA cluster transcription may take advantage of the read-through from a flanking promoter (**Andersen et al., 2017**). Zhang *et al.* (**Zhang et al., 2014**) have shown that tethering Rhino onto a transgene leads to its repression but the production of piRNA depends on the presence of another transgene producing antisense RNA. Moreover, in the context of the *Pld* promoter deletion, a gene flanking the *42AB* piRNA cluster, flies can produce *Pld* piRNAs only if a *Pld* cDNA is expressed in trans (**Andersen et al., 2017**). From all of these observations, a model emerges predicting that simultaneous production of sense and antisense RNA is a shared requirement for piRNA production. However, even if $BX2^{OFF}$ is transcribed on both strands, without additional signals, it still remains inactive for piRNA production.

In addition to having a number of heterochromatic repeats and a double stranded transcription, the production of *de novo* piRNAs from *BX2* requires a triggering signal. From our experiments, *BX2* conversion relies on the simultaneous increase of both sense and antisense RNAs. An active role of euchromatic copies in the establishment of new piRNA clusters by high temperature appears to be consistent with what would naturally happen during the invasion of a naive genome by new TEs or when chromosomal breakages occur leading to the loss of piRNA cluster loci (*Asif-Laidin et al., 2017*). At first, uncontrolled euchromatic TE transposition takes place before the establishment of repression. Such repression would occur after a copy integrates into a preexisting piRNA cluster or by the generation of a new cluster made by successive insertion of nested copies. Consequently, clusters of elements cannot exist without transcriptionally active euchromatic copies. The increase of germline antisense transcripts upon stress or environmental factors, depending on the neighboring genomic environment, and the concomitant presence of numerous sense transcripts from euchromatic active copies, appear to be the starting signals for new piRNA production. These piRNAs can then be inherited at the next generation where they will stably paramutate the corresponding DNA locus with repetitive nature. At that time, the triggering signal is no longer necessary since *BX2* remains activated once flies get back at 25°C. Future generations thus remember what was once considered a threat only through the legacy of maternal piRNAs.

# Materials and methods

## Key resources table

| Reagent type (species) or resource | Designation | Source or reference | Identifiers | Additional information |
|---|---|---|---|---|
| Gene (*Drosophila melanogaster*) | AGO1 | NA | FLYB: FBgn0262739 | |
| Gene (*D. melanogaster*) | RpL32 | NA | FLYB: FBgn0002626 | |
| Gene (*D. melanogaster*) | eEF5 | NA | FLYB: FBgn0285952 | |
| Gene (*D. melanogaster*) | Moonshiner | NA | FLYB: FBgn0030373 | |
| Strain, strain background (*D. melanogaster*) | w1118 | Laboratory Stock | FLYB: FBal0018186 | |
| Strain, strain background (*D. melanogaster*) | BX2 | *Dorer and Henikoff, 1994* PMID:8020105 | FLYB: FBti0016766 | |
| Strain, strain background (*D. melanogaster*) | P(TARGET)[GS] | Bloomington Drosophila Stock Center | FLYB: FBst0011039 | also called P-1039 |
| Strain, strain background (*D. melanogaster*) | P(TARGET)[G] | Bloomington Drosophila Stock Center | FLYB: FBti0003435 | also called BQ16 |
| Strain, strain background (*D. melanogaster*) | P(TARGET)[S] | Bloomington Drosophila Stock Center | FLYB: FBti0003418 | also called BA37 |
| Strain, strain background (*D. melanogaster*) | nosGAL4 | Bloomington Drosophila Stock Center | FLYB: FBti0131635, RRID:BDSC_32180 | |
| Genetic reagent (*D. melanogaster*) | P(lacW) | PMID: 2558049 | FLYB: FBtp0000204 | |
| Genetic reagent (*D. melanogaster*) | P(PZ) | *Mlodzik and Hiromi, 1992* doi: 10.1016/B978-0-12-185267-2.50030–1 | FLYB: FBtp0000210 | |
| Genetic reagent (*D. melanogaster*) | P(A92) | PMID: 2827169 | FLYB: FBtp0000154 | |

*Continued on next page*

*Continued*

| Reagent type (species) or resource | Designation | Source or reference | Identifiers | Additional information |
|---|---|---|---|---|
| Genetic reagent (*D. melanogaster*) | Moon shRNA PA61 | *Andersen et al., 2017* doi:10.1038/nature23482 | | Dr. Julius Brennecke (Institute of Molecular Biotechnology, Vienna) |
| Genetic reagent (*D. melanogaster*) | Moon shRNA PA62 | *Andersen et al., 2017* doi:10.1038/nature23482 | | Dr. Julius Brennecke (Institute of Molecular Biotechnology, Vienna) |
| Antibody | Mouse IgG polyclonal antibody | Merck (ex-Millipore) | Cat# 12-371B, RRID:AB_2617156 | |
| Antibody | Rabbit polyclonal antibody against H3K9me3 | Merck (ex-Millipore) | Cat# 07–442 | |
| Antibody | Rabbit polyclonal antibody against Rhino | PMID: 19732946 | | Dr. William Theurkauf (University of Massachusetts Medical School, Worcester) |
| Sequence-based reagent | RT-qPCR primers | Sigma-Aldrich | | |
| Sequence-based reagent | RT-qPCR primers | Eurogentech | | |
| Commercial assay or kit | RNeasy kit | Qiagen | Cat# 74104 | |
| Commercial assay or kit | Illumina TruSeq Small RNA library preparation kits | Fasteris | http://www.fasteris.com | |
| Commercial assay or kit | Revertaid RT | Thermo Scientific | EP0442 | |
| Commercial assay or kit | Random Hexamers | Invitrogen | N8080127 | |
| Commercial assay or kit | DNaseI (Rnase free) | New Englands Biolabs | M0303S | |
| Commercial assay or kit | dNTPs solution Mix | New Englands Biolabs | N0447S | |
| Commercial assay or kit | Ribolock RNA inhibitor | Thermo Scientific | EO0381 | |
| Commercial assay or kit | Ssofast Evagreen Supermix | Biorad | Cat# 172–5204 | |
| Commercial assay or kit | qPCR kit | Roche | Cat# 04887352001 | |
| Chemical compound, drug | TRIzol | Invitrogen | Cat# 15596026 | |
| Chemical compound, drug | Chloroform | Carlo Erba Reagents | Cat# 438601 | |
| Chemical compound, drug | Chloroform | Sigma-Aldrich | C2432 | |
| Chemical compound, drug | Ethanol (EtOH) | Merck millipore | Cat# 100983 | |
| Chemical compound, drug | Ethanol (EtOH) | Honeywell | Cat# 32221 | |
| Chemical compound, drug | Glycerol | VWR AnalaR NORMAPUR | Cat# 24388.295 | |
| Chemical compound, drug | Glutaraldehyde | Sigma Aldrich | G-5882 | |
| Chemical compound, drug | Potassium hexacyanoferrate(III) | Sigma Aldrich | P3667 | |

*Continued on next page*

*Continued*

| Reagent type (species) or resource | Designation | Source or reference | Identifiers | Additional information |
|---|---|---|---|---|
| Chemical compound, drug | Potassium hexacyanoferrate(II) trihydrate | Sigma Aldrich | P3289 | |
| Chemical compound, drug | X-Gal | Dutscher | Cat# 895014 | |
| Chemical compound, drug | NaCl | VWR AnalaR NORMAPUR | Cat# 27810.295 | |
| Chemical compound, drug | NaCl | Sigma-Aldrich | Cat# 31432 | |
| Chemical compound, drug | Formaldehyde | Sigma-Aldrich | Cat# 252549 | |
| Chemical compound, drug | Schneider Medium | Gibco | Cat# 21720–024 | |
| Chemical compound, drug | Insulin | Sigma-Aldrich | I4011 | |
| Chemical compound, drug | PBS | Ambion | AM9625 | |
| Chemical compound, drug | Triton | Sigma-Aldrich | T8787 | |
| Chemical compound, drug | KCl | Ambion | AM9640G | |
| Chemical compound, drug | HEPES | Fisher Scientific | BP299 | |
| Chemical compound, drug | IPEGAL | Sigma-Aldrich | Cat# 18896 | |
| Chemical compound, drug | DTT | Fisher Scientific | R0861 | |
| Chemical compound, drug | Na Butyrate | Sigma-Aldrich | Cat# 07596 | |
| Chemical compound, drug | EDTA free protease inhibitor | Roche | Cat# 04693159001 | |
| Chemical compound, drug | N lauryl sarkosyl | Sigma-Aldrich | L-5125 | |
| Chemical compound, drug | BSA | Fisher Scientific | BP9703 | |
| Chemical compound, drug | SDS 20% | Euromedex | EU0660-B | |
| Chemical compound, drug | Tris HCl | Invitrogen | Cat# 15504–020 | |
| Chemical compound, drug | Dynabeads A | Invitrogen | 10002D | |
| Chemical compound, drug | Glycine | Sigma-Aldrich | G8898 | |
| Chemical compound, drug | Isopropanol | VWR | Cat# 20842.298 | |
| Software, algorithm | Galaxy Server | ARTBIO | https://mississippi.snv.jussieu.fr/ | |
| Software, algorithm | Weblogo | *Crooks et al., 2004* doi:10.1101/gr.849004 | | |

## Transgenes and strains

All transgenes are in the $w^{1118}$ background. The *BX2* line carries seven *P-lacZ-white* transgenes, (*P (lacW)*, FBtp0000204) inserted in tandem and in the same orientation at cytological site 50C on the

second chromosome (*Dorer and Henikoff, 1994*). The transgene insertion site is located in an intron of the *AGO1* gene (*de Vanssay et al., 2012*). Homozygous individuals are rare and sterile and the stock is maintained in heterozygous state with a *Cy*-marked balancer chromosome. ß-Galactosidase activity from these transgenes cannot be detected in the germline. *P(TARGET)^{GS}* corresponds to *P (PZ)* (FBtp0000210), a *P-lacZ-rosy* enhancer-trap transgene inserted into the *eEF5* gene at 60B7 and expressing ß-Galactosidase in the germline and somatic cells of the female gonads (Bloomington stock number *11039* (FBst0011039). Homozygous flies are not viable and the stock is maintained over a *Cy*-marked balancer chromosome. *P(A92)* (FBtp0000154) is another *P-lacZ-rosy* enhancer-trap transgene that has been used in this study: *P(TARGET)^{G}* corresponds to *BQ16* (FBti0003435) expressing *lacZ* only in the germline and *P(TARGET)^{S}* corresponds to *BA37* (FBti0003418) expressing *lacZ* only in the somatic follicle cells that surround the nurse cells. In both lines, homozygous flies are viable. The *nosGAL4* transgene used is from the *w[*]; PBac{w[+mW.hs]=GreenEye.nosGAL4}Dmel6* line (FBti0131635). Modified miRNA against *moonshiner* (lines *PA61* and *PA62*) were a kind gift from Julius Brennecke (*Andersen et al., 2017*). Additional information about stocks are available at Flybase: 'http://flybase.bio.indiana.edu/".

## Thermic and osmotic treatments

Since maintaining flies at high temperature (29°C) decreases viability, we used the following procedure at each generation: fertilized adult females (G0) were allowed to lay eggs for three days at 25°C on standard cornmeal medium. Adults were then discarded or tested for ovarian ß-Galactosidase expression. Vials containing progeny were transferred at 29°C for the rest of the development until complete emergence of G1 adults. Young adults were then transferred into a new vial where they were allowed to lay eggs for three days at 25°C. For heat-shock treatment, embryos (0–2 hr) were incubated at 37°C for 1 hr as described in *Seong et al. (2011)* and then transferred at 25°C until adult emergence. For osmotic treatments, culturing flies on 300 mM NaCl, as described in *Seong et al. (2011)*, leads to either a large increased time of development or lethality and did not allow us to perform conversion measurements. Accordingly, flies were incubated on 150 mM NaCl for one generation before dissection and ß-Galactosidase staining.

## ß-Galactosidase staining

Ovarian *lacZ* expression assays were carried out using X-gal (5-bromo-4-chloro-3-indolyl-beta-D-galactopyranoside) overnight staining at 37°C as previously described (*Lemaitre et al., 1993*), except that ovaries were fixed afterwards for 10 min. After mounting in glycerol/ethanol (50/50), the germline *lacZ* repression was then calculated by dividing the number of repressed egg chambers by the total number of egg chambers. Most of the time, the total number of egg chambers was estimated by multiplying the number of mounted ovaries by 60, corresponding to an average of three to four egg chambers per ovariole and 16 to 18 ovarioles per ovary. Images were acquired with an Axio-ApoTome (Zeiss) and ZEN2 software.

## Fly dissection and RNA extraction

For each genotype tested, 20 pairs of ovaries were manually dissected in 1X PBS. For small RNA sequencing, total RNA was extracted using TRIzol (Life Technologies) as described in the reagent manual (http://tools.lifetechnologies.com/content/sfs/manuals/trizol_reagent.pdf). For the RNA precipitation step, 100% ethanol was used instead of isopropanol. For RT-qPCR experiments, total RNA was extracted using TRIzol for *BX2* and *w^{1118}* females or RNeasy kit (Qiagen) for *P(TARGET)* females. Up to six biological replicates were used for each genotype.

## Small RNA sequencing analyses

A small RNA fraction of 18 nt to 30 nt in length was obtained following separation of total RNA extracted from dissected ovaries on a denaturing polyacrylamide gel. This fraction was used to generate multiplexed libraries with Illumina TruSeq Small RNA library preparation kits (RS-200–0012, RS200-0024, RS-200–036 or RS-200–048) at *Fasteris* (http://www.fasteris.com). A house protocol based on TruSeq, which reduces 2S RNA (30 nt) contamination in the final library, was performed. Libraries were sequenced using Illumina HiSeq 2000 and 2500. Sequence reads in fastq format were trimmed from the adapter sequence 5'-TGGAATTCTCGGGTGCCAAG-3' and matched to the *D.*

*melanogaster* genome release 5.49 using Bowtie (*Langmead et al., 2009*). Only 18–29 nt reads matching the reference sequences with 0 or one mismatch were retained for subsequent analyses. For global annotation of the libraries (*Supplementary files 5* and *7*), we used the release 5.49 of fasta reference files available in Flybase, including transposon sequences (dmel-all-transposon_r5.49. fasta) and the release 20 of miRNA sequences from miRBase (http://www.mirbase.org/).

Sequence length distributions, small RNA mapping and small RNA overlap signatures were generated from bowtie alignments using Python and R (http://www.r- project.org/) scripts, which were wrapped and run in Galaxy instance publicly from ARTbio platform available at http://mississippi.fr. Tools and workflows used in this study may be downloaded from this Galaxy instance. For library comparisons, read counts were normalized to one million miRNA (*Supplementary files 5* and *7*). A second normalization, performed using the total number of small RNAs matching the *D. melanogaster* genome (release 5.49), gave similar results (*Supplementary files 5* and *7*). For small RNA mapping (*Figures 2* and *4*, *Figure 2—figure supplements 1* and *2*), we took into account only 23–29 nt RNA reads that uniquely aligned to reference sequences. Logos were calculated using Weblogo (*Crooks et al., 2004*) from 3' trimmed reads (23 nt long) matching either *P(lacW)* (*Figure 2B*) or *42AB* (*Figure 2—figure supplement 1B*). The percentage of reads containing a 'U' at the first position was calculated with all 23–29 nt RNA matching the reference sequence (*BX2* transgene in *Figure 2B* and *42AB* in *Figure 2—figure supplement 1B*). Distributions of piRNA overlaps (ping-pong signatures, *Figure 2C* and *Figure 2—figure supplement 1C*) were computed as first described in *Klattenhoff et al. (2009)* and detailed in *Antoniewski (2014)*. Thus, for each sequencing dataset, we collected all of the 23–29 nt RNA reads matching *P(lacW)* or the *42AB* locus whose 5' ends overlapped with another 23–29 nt RNA read on the opposite strand. Then, for each possible overlap of 1 to 29 nt, the number of read pairs was counted. To plot the overlap signatures, a z-score was calculated by computing, for each overlap of 1 to i nucleotides, the number $O(i)$ of read pairs and converting the value using the formula $z(i) = (O(i)-mean(O))/standard\ deviation\ (O)$. The percentage of reads containing a 'A' at the tenth position was calculated within the paired 23–29 nt RNA matching the reference sequence as described in *de Vanssay et al. (2012)* (*BX2* transgene in *Figure 2C* and *42AB* in *Figure 2—figure supplement 1B*). GRH49 (*BX2\**) was previously analyzed in *Hermant et al. (2015)*. Small RNA sequences and project have been deposited at the GEO under accession number GSE116122.

## ChIP experiments

100 ovaries were dissected in Schneider medium supplemented with insulin at room temperature. Cross-linking was performed for 10 min at room temperature in 1X PBS 1% formaldehyde (Sigma). The cross-linking reaction was stopped by adding glycine to a final concentration of 0.125 mM in PBS 0.1% Triton and incubating 5 min on ice. The cross-linked ovaries were washed with 1 ml of PBS 0.1% Triton and crushed in a dounce A potter 20 times. Then a centrifugation at 400 g for 1 min was performed. The pellet was suspended with 1 ml of cell lysis buffer (KCl 0.085 M, HEPES 5 mM, IGE-PAL 0.5%, DTT 0.5 mM, Na butyrate 10 mM, 0.01 M EDTA free protease inhibitor cocktail Roche) and crushed in a dounce B potter 20 times, then 2 mL of cell lysis buffer were added. Centrifugation at 2000 g for 5 min was performed and the pellet was resuspended in 0.5 mL nucleus buffer (HEPES 0.05 M, EDTA 0.01 M, N lauryl Sarcosyl 0.5%, Na butyrate 0.01 M EDTA free protease inhibitor cocktail Roche) and incubated 15 min in a cold room on a rotator. Sonication was performed with Bioruptor (Diagenode) set to high power for 10 cycles (15 s on and 15 s off). A centrifugation was performed 15 min at 16000 g at 4°C. Five μg of chromatin was used for each immunoprecipitation. A preclear of 4 hr was performed with 25 μL of dynabead Protein A. The immunoprecipitation reaction was performed with 50 μL of dynabead Protein A coated with 5 μg of antibodies (H3K9me3 polyclonal antibody C1540030 diagenode or Normal Mouse IgG polyclonal antibody 12–371 Millipore), or 20 μL of serum for the Rhino antibody (kindly provided by Dr W. Theurkauf) over night in the cold room on a rotator. Three washes of 10 min in a high salt buffer (Tris HCl pH 7.5 0.05 M, NaCl 0.5 M, Triton 0.25%, IGEPAL 0.5%, BSA 0.5%, EDTA 5 mM) were performed and the elution of chromatin was performed 30 min with 500 μL of elution buffer (Tris pH 7.5 0.05 M, NaCl 0.05 M, EDTA 5 mM, SDS 1%); RNase treatment was omitted; H3K9me3 and Rhino ChIP were respectively done on 5 and 4 independent biological samples followed by qPCR (Roche light Cycler) on each sample. Values were normalized to respective inputs and to two genomic regions known to be enriched in H3K9me3 and Rhino (*42AB*): region 1 (chr2R: 6449409–6449518) and region 2 (chr2R:

6288809–6288940). An unpaired *t*-test was used to calculate significance of the differences (p<0.05). Error bars represent the standard deviation.

## RT-qPCR experiments

For each sample, 10 µg of total RNA was treated with DNase (Fermentas). For classical RT-qPCR experiments, 1 µg of DNase-treated RNA was used for reverse transcription using random hexamer primers (Fermentas). Real-time qPCR was performed on triplicates of each sample. *RpL32* was used as reference. The same series of dilution of a mix of different RT preparations was used to normalize the quantity of transcripts in all RT preparations leading to standard quantity (Sq) values. Variations between technical triplicates was very low when compared to variations between biological replicates. The mean of the three technical replicates was then systematically used (meanSq). For each biological sample, we calculated the ratio meanSq(gene)/meanSq(*RpL32*) to normalize the transcript quantity. Then, the mean of each sample ratio was used to compare the two conditions. For sense-specific RT-qPCR experiments, three reverse transcription were performed using 1 µg of DNase-treated RNA (Fermentas): first without primer (control RT), second with a *lacZ* sense primer (anti-sense transcript specific RT) and third with a *lacZ* antisense primer (sense transcript specific RT). qPCR was then performed on technical triplicates of each RT using a primer pair specific for *lacZ* sequence. A series of dilutions - ranging from $50 \times 10^{-15}$ g.µl$^{-1}$ to $0.08 \times 10^{-15}$ g.µl$^{-1}$ - of a plasmid containing the *P(lacW)* transgene was used as reference to normalize the quantity of *lacZ* transcripts (Sq values). The number of molecules was estimated by considering that *P(lacW)* is 11191 bp long and that the average weight of a base pair is 650 g/mol. Using Avogadro's number, the number of copies was estimated as equal to the dsDNA amount (in g) times $6.022 \times 10^{23}$ divided by the dsDNA length times 650. For example, $50 \times 10^{-15}$ g corresponds to approximately 4139 molecules. Variations between technical triplicates were very low when compared to variations between biological replicates. The mean of the three technical replicates was then systematically used (meanSq). The measure of the quantity of transcripts (sense or antisense) for a biological sample was then calculated as the (meanSq(sense or antisense specific) - meanSq(control)). This allowed us to eliminate background noise due to unspecific RT amplification for both sense or antisense without specific primer. The mean of each sample ratio was used to compare the two conditions.

## Primer sequences

For classical RT-qPCR experiments, primers used were for *w* (ChIP experiment): 5'-GTCAATG TCCGCCTTCAGTT-3' and 5'-GGAGTTTTGGCACAGCACTT-3', these primers are specific of the *P (lacW)* transgene in a *w^1118* background; for *42AB* regions, 5'-TGGAGTTTGGTGCAGAAGC-3' and 5'-AGCCGTGCTTTATGCTTTACT-3' (region 1) and 5'-AAGACCCAATTTTTGCGTCGC-3' and 5'-CAAGGATAGGGATTTGGTCC-3' (region 2); for *RpL32*: 5'-CCGCTTCAAGGGACAGTATCTG-3' and 5'-ATCTCGCCGCAGTAAACGC-3'; for *lacZ*: 5'-GAGAATCCGACGGGTTGTTA-3' and 5'-AAATTCA-GACGGCAAACGAC-3'; for *eEF5*: 5'-TAACATGGATGTGCCCAATG-3' and 5'-AACGCAATTG TTCACCCAAT-3'; for *AGO1*, primers have been chosen in order to detect spliced forms of transcripts coming upstream of the insertion point of *BX2* and encoding AGO1-RA, -RD and -RC isoforms (*Figure 4C*): 5'-GGATCTCCAGATGACCTCCA-3' and 5'-GGACACTTGTCCGGCTGTAT-3'. For detecting all *AGO1* transcripts isoforms, including the AGO1-RB isoform that originates from a promoter located downstream the *BX2* insertion point: 5'-ATGAGCCGGTCATCTTTTTG-3' and 5'-GGCAATCGATGGTTTCTTGT-3'. For sense-specific RT-qPCR experiments, we used specific primers during the reverse transcription step: 5'-AGTACGAAATGCGTCGTTTAGAGC-3' for detection of antisense *lacZ* transcripts and 5'-AATGCGCTCAGGTCAAATTC-3' for detection of sense *lacZ* transcripts.

## Acknowledgements

We thank Doug Dorer, Steve Henikoff, Julius Brennecke and the *Bloomington Stock Center* for providing stocks. We thank Bill Theurkauf for providing antibodies. We thank flybase.org for providing databases. We thank Ritha Zamy for technical assistance. We thank Clément Carré, Lori Pile, Ana Maria Vallès and Jean-René Huynh for critical reading of the manuscript. We thank Christophe Antoniewski for helpful advices and development of the ARTbio platform (http://artbio.fr/).

## Additional information

### Funding

| Funder | Grant reference number | Author |
|--------|------------------------|--------|
| Fondation ARC pour la Recherche sur le Cancer | SFI20131200470 | Stéphane Ronsseray |
| Fondation pour la Recherche Médicale | DEP20131128532 | Stéphane Ronsseray |
| Agence Nationale de la Recherche | plastisipi | Stéphane Ronsseray |
| Pierre and Marie Curie University | EME1223 | Laure Teysset |

The funders had no role in study design, data collection and interpretation, or the decision to submit the work for publication.

### Author contributions

Karine Casier, Conceptualization, Data curation, Formal analysis, Validation, Investigation, Visualization, Methodology, Writing—review and editing; Valérie Delmarre, Data curation, Validation, Investigation; Nathalie Gueguen, Catherine Hermant, Elise Viodé, Validation, Investigation; Chantal Vaury, Funding acquisition, Writing—review and editing; Stéphane Ronsseray, Conceptualization, Resources, Funding acquisition, Investigation, Writing—review and editing; Emilie Brasset, Validation, Investigation, Writing—review and editing; Laure Teysset, Supervision, Funding acquisition, Investigation, Visualization, Project administration, Writing—review and editing; Antoine Boivin, Conceptualization, Data curation, Formal analysis, Supervision, Validation, Investigation, Visualization, Methodology, Writing—original draft, Project administration, Writing—review and editing

### Author ORCIDs

Karine Casier http://orcid.org/0000-0001-6825-8057
Elise Viodé https://orcid.org/0000-0002-5058-2710
Laure Teysset https://orcid.org/0000-0002-7413-1850
Antoine Boivin https://orcid.org/0000-0001-8671-1599

### Decision letter and Author response
Decision letter https://doi.org/10.7554/eLife.39842.032
Author response https://doi.org/10.7554/eLife.39842.033

## Additional files

### Supplementary files

• Supplementary file 1 .Silencing capacities of $BX2^{ON}$ and $BX2^{OFF}$ lines throughout generations at 25°C and at 29°C. $BX2^{OFF}$ and $BX2^{ON}$ are recombined lines carrying the $P(TARGET)^{GS}$ and the $BX2$ locus transgenes on the same chromosome. Numbers show the fraction of females harboring complete germline repression of $P(TARGET)^{GS}$ at each generation. Complete stability of the initial epigenetic state was observed at 25°C for $BX2^{OFF}$ and $BX2^{ON}$ lines, 0% repression (n = 415) and 100% repression (n = 339), respectively. At 29°C, all $BX2^{OFF}$ lines showed emergence of silencing capacities, 24.7% (n = 3812). $BX2^{ON}$ lines maintained their silencing capacities over generations at 29°C, 100% (n = 377). nt: not tested.
DOI: https://doi.org/10.7554/eLife.39842.017

• Supplementary file 2. Stability of $BX2^{\Theta}$ lines. Five $BX2, P(TARGET)^{GS}$ lines showing full repression capacities after 23 generations kept at 29°C were transferred at 25°C and tested for their silencing capacities throughout generations. Numbers show females presenting full $P(TARGET)^{GS}$ repression and the total number of tested flies. In all cases, the $BX2^{ON}$ epiallele induced by high temperature remains completely stable during 50 additional generations at 25°C.

DOI: https://doi.org/10.7554/eLife.39842.018

• Supplementary file 3. Maternal effect of $BX2^{ON}$ lines. Reciprocal crosses were performed at 25°C between $BX2^{ON}$, $P(TARGET)^{GS}$ ($BX2^{\Theta}$ or $BX2*$) individuals and flies carrying a balancer of the second chromosome ($Cy$). For both the maternal and paternal inheritances (named MI and PI, respectively), lines were established and maintained at 25°C by crossing G1 individuals carrying the $BX2^{\Theta}$, $P(TARGET)^{GS}$ chromosome over $Cy$ (**Figure 1—figure supplement 2A**). Silencing capacities of these lines was tested over generations by intra-strain ovarian ß-Galactosidase staining. Numbers represent the fraction of females showing complete repression of $P(TARGET)^{GS}$. In all cases, maternal transmission of the $BX2^{\Theta}$ cluster results in progeny showing complete repression capacities which are stable over generations whereas paternal transmission of the $BX2^{\Theta}$ cluster results in the definitive loss of $BX2$ silencing capacities similarly to the $BX2*$ epigenetic state.

DOI: https://doi.org/10.7554/eLife.39842.019

• Supplementary file 4. Paramutagenic effect of $BX2^{ON}$ lines. The capacity of the cytoplasm of $BX2^{ON}$ females to activate a $BX2^{OFF}$ cluster was tested as shown in the mating scheme. $BX2^{ON}$ females (either $BX2^{\Theta}$ or $BX2*$) were crossed with $BX2^{OFF}$ males, incubated at 25°C. Lines were established with G1 individuals, which have maternally inherited piRNAs and paternally inheritance of the $BX2$ cluster (**Figure 1—figure supplement 2B**). These lines were maintained at 25°C and their silencing capacities were tested over generations by crossing females with $P(TARGET)^{GS}$ males. Numbers represent the fraction of females showing complete repression of $P(TARGET)^{GS}$. All of the derived lines showed complete silencing capacities over generations, revealing that the cytoplasm of $BX2^{ON}$ females, either $BX2^{\Theta}$ or $BX2*$, can fully activate a $BX2^{OFF}$ cluster.

DOI: https://doi.org/10.7554/eLife.39842.020

• Supplementary file 5. Annotation of small RNA libraries. Small RNAs were prepared from ovaries of females of the indicated genotype. Values for the different categories of sequences are the total number of sequence reads that matched reference libraries. For comparisons, libraries were normalized (normalization factor) to 1 million miRNA (miRNA rpm) or to 1 million Dmel reads (Dmel rpm).

DOI: https://doi.org/10.7554/eLife.39842.021

• Supplementary file 6. Silencing capacities of $BX2^{ON}$ and $BX2^{OFF}$ lines across generations cultured at 25°C and at 29°C. Same as **Supplementary file 1** except that egg chambers were monitored for $P(TARGET)^{GS}$ repression instead of whole ovaries. Numbers show the fraction of repressed egg chamber per generation.

DOI: https://doi.org/10.7554/eLife.39842.022

• Supplementary file 7. Annotation of small RNA libraries from $BX2^{OFF}$ raised at 25°C or 29°C. Small RNAs were prepared from ovaries of females of the indicated genotype. Values for the different categories of sequences are the total number of sequence reads that matched reference libraries. For comparisons, libraries were normalized (normalization factor) to 1 million miRNA (miRNA rpm) or to 1 million Dmel reads (Dmel rpm).

DOI: https://doi.org/10.7554/eLife.39842.023

• Supplementary file 8. $P(TARGET)^{GS}$ requirement in the $BX2$ conversion process. Comparison of the conversion frequency in one generation between $BX2$, $P(TARGET)^{GS}$ (**Figure 3**) and $BX2$ (**Figure 3—figure supplement 1**) genotypes. The difference between the presence and absence of the $P(TARGET)^{GS}$ transgene is highly significant ($p=8.5\times10^{-6}$, homogeneity $\chi^2 = 23.35$ with 2 degrees of freedom).

DOI: https://doi.org/10.7554/eLife.39842.024

• Supplementary file 9. Silencing capacities of $BX2^{OFF}$ lines recombined in a $P(TARGET)^{GS}$ background throughout generations developed at 29°C. $BX2^{OFF}$ was initially recombined with a line carrying the $P(TARGET)^{GS}$ transgene to obtain the $BX2^{OFF}$, $P(TARGET)^{GS}$ lines. From these crosses, eight independent recombinants without the $P(TARGET)^{GS}$ transgene were recovered and were further cultured at 29°C. To test if some of them acquired silencing capacities, females were crossed with males harboring the $P(TARGET)^{GS}$ transgene and their progeny was stained for ß-Galactosidase expression. Numbers show the fraction of females harboring complete germline repression of $P(TARGET)^{GS}$ at each generation. A complete stability of the initial epigenetic OFF state was observed for all recombinant lines.

DOI: https://doi.org/10.7554/eLife.39842.025

• Supplementary file 10. Silencing capacities of $BX2^{OFF}$; $P(TARGET)^G$ lines throughout generations at 25°C and at 29°C. Numbers show the fraction of females harboring complete germline repression of $P(TARGET)^G$ at each generation. Complete stability of the initial epigenetic state was observed at 25°C for $BX2^{OFF}$, 0% repression (n = 189). At 29°C, $BX2^{OFF}$; $P(TARGET)^G$ lines showed emergence of silencing capacities, 19.79% (n = 6766).
DOI: https://doi.org/10.7554/eLife.39842.026

• Supplementary file 11. Heat shock and saline stresses do not induce conversion of $BX2^{OFF}$. $BX2^{OFF}$, $P(TARGET)^{GS}$ flies were raised during one generation either on classical cornmeal medium (control) at 25°C or were heat shocked for 1 hr at 37°C at the 0–2 hr embryo stage or were cultured on medium supplemented with 150 mM NaCl. The female progeny were then stained for ß-Galactosidase expression and egg chambers were individually monitored in order to detect any possible conversion event. No repressed egg chambers were observed (0/total number of egg chambers). Compared to results obtained at 29°C (from data observed in G1 in *Figure 3*), differences are significant: for heat shock experiment, p=$7.2{\times}10^{-25}$, homogeneity $\chi^2$ = 106.03 with 2 degrees of freedom and for NaCl experiment, p=$4.9{\times}10^{-27}$, homogeneity $\chi^2$ = 115.93 with 2 degrees of freedom.
DOI: https://doi.org/10.7554/eLife.39842.027

• Transparent reporting form
DOI: https://doi.org/10.7554/eLife.39842.028

## Data availability

Small RNA sequences and project have been deposited at the GEO under accession number GSE116122.

The following dataset was generated:

| Author(s) | Year | Dataset title | Dataset URL | Database and Identifier |
|---|---|---|---|---|
| Casier K, Boivin A | 2018 | Environmentally-induced epigenetic conversion of a piRNA cluster | https://www.ncbi.nlm.nih.gov/geo/query/acc.cgi?&acc=GSE116122 | NCBI Gene Expression Omnibus, GSE116122 |

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
