## [Decision Letter]

[**Editorial note:** This article has been through an editorial process in which the authors decide how to respond to the issues raised during peer review. The Reviewing Editor's assessment is that all the issues have been addressed.]

Thank you for submitting your article "Environmentally-induced epigenetic conversion of a piRNA cluster" for consideration by *eLife*. Your article has been reviewed by three peer reviewers, and the evaluation has been overseen by a Reviewing Editor and James Manley as the Senior Editor. The reviewers have opted to remain anonymous.

We have included the separate reviews below for your consideration. We would recommend that you first respond to outline your replies to the reviewers and what revisions you intend to make, so that we can offer further guidance. We note that all the reviewers felt that more mechanistic insight would be desirable. If you have any questions, please do not hesitate to contact us.

Separate reviews (please respond to each point):

*Reviewer #1:*

The authors report how high temperature (environment) can induce a DNA stretch to become a piRNA-producing locus in female flies.

Findings

1) Some (25% of the females studied) flies within a population are randomly induced to turn an inactive piece of DNA into an active locus that generates piRNAs, when grown at higher temperature (29^o^C instead of normal 25^o^C). This is monitored by the clever use of a b-gal reporter assay. This induction increases with generation time.

2) Such temperature-activated clusters are identical to those activated by maternally-inherited piRNAs. They present similar piRNAs produced and are decorated with the expected chromatin and piRNA-factor marks.

3) The higher temperature increased antisense RNA production from the previously-silent locus and this was channelled into the siRNA pathway and did not lead to increased piRNA production.

4) The presence of the complementary target RNA (expressed from a euchromatic locus) was essential for the high temperature-triggered piRNA generation.

Concerns

1) Abstract, Second sentence: There is no data to show that piRNA clusters are maternally defined except in flies. In mice germ cells are induced from the somatic lineage at embryonic day 7.5 and there is no possibility of maternal inheritance playing a role. If the authors want to keep this sentence, specify it is for flies.

2) Abstract, third sentence: The sentence sounds very dramatic as it means that higher temperature alone was sufficient to convert the P-transgene locus into a piRNA cluster. Perhaps the features of the genomic region (having P-transgenes), location, chromatin status etc are responsible. I would rephrase it. It is misleading as although temperature might be the trigger, this locus might already be primed and ready to go.

3) The authors make a very striking finding regarding high temperature-induced piRNA production from a locus, and link it to presence of both sense and antisense transcripts are necessary for this. This still leaves open the question how the locus was made into piRNAs. There is no explanation for this.

4) It would be good to have the actual sequence information for the different transgenes and the b-gal reporter, and the actual transcripts present in these flies. One is left in the dark as to the level of complementarity that exists between them when they are referred to as sense and antisense transcripts.

In conclusion, the observations made are striking and extremely interesting, but perhaps a bit more of molecular insights might be useful to appreciate the observations made.

*Reviewer #2:*

Piwi-interacting RNAs (piRNAs) produced from piRNA clusters repress transposable elements in animal gonads. It has been shown that use of piRNA clusters is determined by maternally-deposited piRNAs. In this study, Casier et al. however showed that raising flies at high temperature (29^o^C) resulted in ectopic expression of piRNAs from the *BX2* transgene locus in flies that had no maternally-deposited, *BX2*-derived piRNAs. The *BX2* transgene consists of P5’, lacZ, white, a part of pBR322 (pBR) and P3’, and does not express piRNAs at normal temperature (25^o^C). The authors claimed that this was the first case showing that heat treatment can activate de novo piRNA production from the *BX2* locus (i.e., non-piRNA cluster) in a manner independent of maternally-deposited piRNAs.

The authors' findings also include:

1) Activation of de novo piRNA production by heat treatment was restricted to the *BX2* locus.

2) *BX2*-derived piRNAs produced upon heat treatment showed molecular properties similar to those of *BX2*-derived piRNAs expressed in a manner dependent of maternally-deposited piRNAs.

3) Ectopically expressed lacZ-piRNAs from the *BX2* locus were capable of silencing βGal over generations.

4) The βGal silencing effect was low at the first generation but gradually increased over generations.

5) Rhino and H3K9me3 accumulated at the *BX2* locus upon heat treatment. However, this may not be the cause to ectopic expression of *BX2*-derived piRNAs, as the authors claimed that *BX2*^OFF^ and *BX2*^ON^ produced anyway similar amounts of sense and antisense transcripts (subsection “Epigenetic conversion at 29°C occurs at a low rate from the first generation”).

6) An increase of transcription from the AGO1 promoter at 29^o^C could lead to an increase of *BX2* antisense RNA transcription. The *BX2* transgene was inserted in the AGO1 gene in a convergent transcription manner.

7) Activation of *BX2*-derived piRNA production by heat treatment required a homologous sequence.

I found the authors' finding that heat treatment induces de novo piRNA production from non-piRNA clusters potentially interesting. However, the mechanistic insights remain vague. The authors claimed that homologous sequence was required for the induction. However, to claim this, much more supportive data should be provided. The authors also found that H3K9me3 and Rhino started to accumulate at the *BX2* locus upon heat treatment. However, the meaning of the accumulation remains unclear. My other concerns were indicated below. I hope that this reviewer's concerns might be helpful for the authors to revise the manuscript.

Concerns and suggestions from this reviewer:

1) The authors proposed that the interaction between the excess of *BX2* antisense transcripts and the sense *P(TARGET)* transcripts is a prerequisite for the production of de novo piRNAs (subsection “Heat conversion requires a homologous sequence *in trans*”). To confirm this, I recommend the authors conducting experiments using heat treated *BX2*^OFF^ that do not have lacZ at the *BX2* loci but contains *P(TARGET)*.

2) The level of AGO1 RNA transcripts should be examined in *BX2*^OFF^ line containing *P(TARGET)*. I do not understand why the authors used the line without *P(TARGET)* in this particular experiment.

3) Figure 4B: Upon heat treatment, the expression level of *BX2* antisense transcripts was raised. Was this phenomenon related to Rhino and H3K9me3 accumulation observed in Figure 2DE? Experiments in Figure 2 and Figure 4 should be performed using the same fly lines.

4) Is the *BX2* transgene in the *BX2*^OFF^ line containing *P(TARGET)* also inserted in the AGO1 gene? This should be examined. If this were the case, the authors better examine whether piRNAs were derived from the AGO1 gene.

5) The authors should test experimentally whether *BX2*-derived piRNAs are loaded into PIWI proteins. This is very important.

6) Figure 2B: Explain why *BX2*-derived piRNAs did not show 10A bias. The data shown in Figure 2C supported the idea that *BX2*-derived piRNAs were products of the ping-pong pathway. Then, they should have shown 10A bias, but in reality they did not.

7) Figure 2DE: The authors should examine genome-widely where in the genome H3K9me3 and Rhino accumulated upon heat treatment. It is hard to imagine that H3K9me3 and Rhino accumulation only occurred at the *BX2* locus upon heat treatment.

8) The authors should examine whether H3K9me3 and Rhino accumulated at the *BX2* locus in *BX2*^OFF^ line without *P(TARGET)* upon heat treatment.

9) This manuscript should be edited thoroughly by a native English speaker.

10) Table 1 should be removed from the main text.

*Reviewer #3:*

This work takes advantage of a clever system using suppression of a lacZ reporter gene as a read out indicating the production of piRNAs from the *BX2* locus. Use of this system has resulted in several novel and interesting results, most notably the paramutation phenomenon. In this work, Casier et al. show that the *BX2* locus is silent at 25^o^C, but produces piRNAs at 29 ^o^C in the presence of homologous sequence in the background.

While I have no problem with the experiments, but did wish for more insight into the mechanism, and more caution in the interpretation. In particular, entirely environmental specification of piRNA clusters implied in the Abstract is a strong claim, and as the authors show themselves, is an oversimplification. The induction they find is quite specific to their system. Specifically, there are higher levels of homologous sense and anti-sense transcript at 29 than at 25, and temperature alone doesn't change other piRNA production. Similarly, it would be useful to know if the expression of the sense transcript from P-Target is also higher at 29 that at 25. (I think this construct has a heat-shock promoter, as well as that of the P-element, which shows temperature sensitive effects.)

I wasn't entirely convinced by the suggestion that expression of natural piRNA clusters is higher at 29 ^o^C. Many piRNAs are generated as a secondary byproduct of transposable element message, and many transposable elements seem to have temperature sensitive expression (possibly as a byproduct of changes in chromatin.) The piRNA dependent splicing suppression of the P-element is also, apparently, not temperature sensitive (Teixiera et al., 2017).

Minor points. It does seem important (and quick) to eliminate the possibility that there's any active P-elements in these lines. While *w1118* is a classic M-type background, there are some cases of sublines of other M-types harboring P-elements via contamination (Rahman et al. Nucleic Acids Research, Volume 43, Issue 22, 15 December 2015, Pages 10655-10672). If there is active P-element, I think it is possible that transgenes (which has the TIRs necessary for P-element insertion) has been picked up and put in another piRNA cluster. It's a simple matter of a few PCRs to exclude this possibility. The Cy and CyRoi backgrounds should be tested as well, to show that they are devoid of any plus-strand transcript that could be an ongoing trigger for piRNA production.

Subsection “Epigenetic conversion at 29°C occurs at a low rate from the first generation” paragraph two. The model is entirely sensible, but the data are quite noisy, and seem like they would be consistent with any model predicting an increase over generations. Can they compare this model to, for example, a model without the “c” parameter using Akaike Information Criteria or a log-likelihood test? This would make the analysis more informative.

Subsection “Heat conversion requires a homologous sequence *in trans*”: The chi-square statistic and degrees of freedom should be reported as well as the p-value. (And this is very minor, but the Greek letter is usually used in place of “chi”.)

[Editors' note: further revisions were suggested, as described below.]

Thank you for resubmitting your work entitled "Environmentally-induced epigenetic conversion of a piRNA cluster" for further consideration at *eLife*. Your revised article has been evaluated by James Manley (Senior Editor), a Reviewing Editor, and two reviewers.

The manuscript has been improved but the referees still have significant concerns. If you decide to proceed with publication of the present version, our editorial rating, which will be displayed immediately below the Abstract, would be that "major issues remain unresolved."

The reviewer comments (on the original version and revised version) would also be published, along with your responses.

Alternatively, you might decide to undertake significant extra work to try to address the concerns. Or, you might decide to shorten the paper (highlighting the most interesting findings) and leaving extensive additional work for a future paper. You also have the option to withdraw the present paper from consideration.

The reviews follow below, and we will look forward to hearing how you propose to proceed.

*Reviewer #1:*

In the revised version of the manuscript the authors find an involvement of sense-antisense transcripts and Dicer-2 in piRNA biogenesis. The authors may remember that one of the early findings celebrated as a major discovery was the lack of a role for double-stranded RNAs and the double-stranded RNA cleaving enzyme Dicer, in piRNA biogenesis. If now, the authors find a role for these in piRNA biogenesis, it has to be backed with strong data.

1) Can the authors demonstrate loading of the 21 nt RNAs into Piwi proteins?

2) If its role is epigenetic conversion, then ideally it has be loaded into the nuclear Piwi.

3) Are there experiments linking Dicer-2 to loading into a Piwi protein? Are they ever found in the same complexes?

I agree that there is a strong functional/genetic data that is mysterious, but very interesting. The heat-activated piRNAs are functional as they can silence a LacZ transgene. Since this manuscript will be published in some form, I am happy to support description of this part in a revised manuscript, but without wild speculation of Dicer-2 in piRNA biogenesis (even if they examine a Dicer mutant). These attempts at explaining the molecular mechanism may be toned down. I am worried that any prominent mention of Dicer-2 in this context may only serve to foster suspicion of the interesting genetic observation.

*Reviewer #3:*

I still find this study clever and interesting, and think the manuscript shows some specific improvements:

The “primed” nature of the *BX2* locus is clearer, and the claim in the paper is more measured and reasonably justified.

The experiment showing the Dcr2 dependence of the conversion of the *BX2* from off to on goes a little way toward understanding the mechanism of piRNA cluster formation, implying that it is dependent on siRNAs.

The addition of the figure showing the model for the mechanism. The weak point of the proposed mechanism, as I see it, is the lack of evidence that the siRNAs are loaded onto PIWI, any evidence the authors can provide for this mechanism would greatly improve the paper. I can't make any suggestions that seem technically feasible, however, in light of the low conversion rate in this system.

Further clarification would also be helpful for these aspects:

Regarding the model in Figure 1—figure supplement 3: I previously suggested a statistical analysis of this model. Rather than the more detailed analysis I suggested previously, I think a reasonable compromise would be to show that there is a significant increase over generations. That is the important point here (and, in fact, it's hard to imagine a scenario where the “memory” of conversion does not play a role in the increase over generations). But, while there appears to be a trend, the data are sufficiently noisy that it would be useful just to see that the increase is significant (perhaps a linear regression on transformed data?).

Discussion opening paragraph: Please explain what you mean by the “specific spliced transcript” comment; it wasn't clear.

e.g., supplementary file 11: It would be nice to more cautious in the interpretation where there are large differences in the number of flies examined. In Figure 3, for example, I think the numbers show that 41 of 1447 females show partial or complete repression; the comparable (?) numbers is Supplementary file 11 show 0 of 32 (?) females show any repression. This is not a significant difference via Fisher's exact test.

Additional data files and statistical comments:

Please see major comments for suggestions regarding statistical analyses, particularly in cases where conclusions are drawn about the absence of repression in small sample sizes.

---

## [Author Response]

Reviewer #1:

[…] Concerns1) Abstract, Second sentence: There is no data to show that piRNA clusters are maternally defined except in flies. In mice germ cells are induced from the somatic lineage at embryonic day 7.5 and there is no possibility of maternal inheritance playing a role. If the authors want to keep this sentence, specify it is for flies.

We have specified that the hypothesis of the determination of piRNA cluster by maternal inheritance of homologous piRNA is for flies.

2) Abstract, third sentence: The sentence sounds very dramatic as it means that higher temperature alone was sufficient to convert the P-transgene locus into a piRNA cluster. Perhaps the features of the genomic region (having P-transgenes), location, chromatin status etc are responsible. I would rephrase it. It is misleading as although temperature might be the trigger, this locus might already be primed and ready to go.

The reviewer #1 is completely right, our results show that the locus is ready to become a piRNA cluster and the temperature is the trigger for the activation. We have rephrased the sentence.

3) The authors make a very striking finding regarding high temperature-induced piRNA production from a locus, and link it to presence of both sense and antisense transcripts are necessary for this. This still leaves open the question how the locus was made into piRNAs. There is no explanation for this.

This is the major point raised by the three reviewers. The major concern in our system is the low occurrence of the epigenetic conversion (2%). It was possible to identify and quantify it through generations because of the use of the high reliability of the ßGalactosidase staining allowing us to (i) visualize the status of each eggs chambers individually at each generation, (2) study a large number of flies (Figure 1 and 3, for instance), (3) do statistics. However, observing 2% difference during the process of activation by molecular analysis appeared impossible (for example, the observation of 2% increase of piRNAs or chromatin modification through ChIP analyses is not statistically possible). Therefore, molecular analyses were performed on stable epigenetic states (*BX2^OFF^* or *BX2^ON^* converted by maternally inherited piRNAs or by high temperature) and after being transferred at 25°C.

In order to answer to this major point, we performed additional experiments to test if double stranded RNAs should be formed prior piRNA biogenesis to activate *BX2*. So, we tested whether *Dcr-2* might affect the activation of *BX2* by temperature. We generated flies containing *BX2, Dcr-2* mutant allele and a *P(TARGET)* tested the *BX2* activation at each generation by ßGalactosidase staining. We present results showing that *Dcr-2* mutation impairs the activation of *BX2* by high temperature (Figure 3—figure supplement 3). This result is consistent with the requirement of double strand RNA intermediates for *BX2* conversion into a piRNA cluster. We have added these experiments at the end of the result section as well as an additional figure (Figure 5) containing a cartoon putting together all of the results in a model. We hope that these new data will satisfy the reviewer concerns on the mechanism of activation.

4) It would be good to have the actual sequence information for the different transgenes and the b-gal reporter, and the actual transcripts present in these flies. One is left in the dark as to the level of complementarity that exists between them when they are referred to as sense and antisense transcripts.

We have added the precise structure of all transgenes used in this study showing that the complementarity between the *BX2* and the *P(TARGET)* transgenes is restricted to the *lacZ* gene and the 5' and 3' region of the *P* element required for transgenesis (new Figure 1—figure supplement 1).We have specified what are sense and antisense *BX2* transcripts (they correspond to sense and antisense *lacZ* transcripts, respectively) in the text and in the legend of the Figure 4.

In conclusion, the observations made are striking and extremely interesting, but perhaps a bit more of molecular insights might be useful to appreciate the observations made.

We appreciate the interest of this reviewer to our work and hope that the additional genetic results obtained with others euchromatic transgenes (see below) and with *Dcr-2* sufficiently improve the molecular understanding of the phenomenon we are describing.

Reviewer #2:

*[…] I found the authors' finding that heat treatment induces* de novo *piRNA production from non-piRNA clusters potentially interesting. However, the mechanistic insights remain vague.*

As previously described, we hope that the experiments with others transgenes (see below) and with *Dcr-2* give more insight on the mechanistic part of the phenomenon. Based on the new results presented in this revised version, we have added a new figure presenting a model of mechanism (Figure 5).

The authors claimed that homologous sequence was required for the induction. However, to claim this, much more supportive data should be provided.

We have performed additional experiments using another target, a *P-lacZ-rosy* reporter transgene inserted in euchromatin and expressed in the germline (*P(A92), BQ16*, FBti0003435). In the presence of this transgene, *BX2* was converted at 29°C at the same rate than with *P(TARGET)* showing that *BX2* conversion of *BX2* by temperature cannot be attributed to any specificity linked to the *P(TARGET)* insertion. We have added these results in the main text and in a supplementary table (Supplementary file 10).Moreover, the potential effect of the background has been now tested over 30 generations and no conversion event has been observed, reinforcing the specific role of the germline expressed target in the activation of *BX2*. The Supplementary file 9 has been completed. We also performed a new experiment using a *P(A92)* transgene that is expressed in the somatic follicle cells but not in the germline nurse cells. With this transgene, no conversion was observed, showing that the euchromatic homologous transgene need to be transcribed (Figure 3—figure supplement 2). This reinforces the role of the "sense" transcripts brought by the euchromatic transgene in the conversion process.

The authors also found that H3K9me3 and Rhino started to accumulate at the BX2 locus upon heat treatment. However, the meaning of the accumulation remains unclear. My other concerns were indicated below. I hope that this reviewer's concerns might be helpful for the authors to revise the manuscript.

As a matter of fact, the enrichment in H3K9me3 and Rhino on the *BX2* locus can be interpreted by a progressive accumulation of all the loci present in egg chambers studied or by a progressive increase number of egg chambers activated through generation (each one of them being fully activated or not activated at all, ON/OFF conversion). Based on our results obtained by *lacZ* staining showing exclusively egg chambers converted or not converted with low occurrence, we favor an on/off conversion per egg chamber induced by maternal inheritance of homologous piRNA or by temperature.

Concerns and suggestions from this reviewer:

*1) The authors proposed that the interaction between the excess of BX2 antisense transcripts and the sense P(TARGET) transcripts is a prerequisite for the production of* de novo *piRNAs (subsection “Heat conversion requires a homologous sequence in trans”). To confirm this, I recommend the authors conducting experiments using heat treated BX2^OFF^ that do not have lacZ at the BX2 loci but contains P(TARGET).*

This could be a good idea but we cannot modify at will the *BX2* locus that had been obtained through several rounds of transposition followed by thousand of flies selection (Dorer and Henik_off_ 1994). *P*-transgenes have the capacity to inserted in the genome without hot spots (except gene regions rich), therefore it seems to be very difficult to recover 7 insertions within the same location. Moreover, if 7 *P(lacW)* was identified in another locus, it will not have necessary the same biological properties than the *BX2* locus.

2) The level of AGO1 RNA transcripts should be examined in BX2^OFF^ line containing P(TARGET). I do not understand why the authors used the line without P(TARGET) in this particular experiment.

The presence of the *BX2* cluster may affect specific transcripts of one allele of the *AGO1* gene and to be sure to measure the effect of temperature on *AGO1* transcripts, we performed experiments without *BX2*. However, we have reproduced the measurement of *AGO1* steady-state RNA level by qRT-PCR experiment in a *BX2* context, with and without *P(TARGET)* and confirmed that the amount of *AGO1* transcripts is higher at 29°C compared to 25°C. These results have been added in the Figure 2—figure supplement 2.

3) Figure 4B: Upon heat treatment, the expression level of BX2 antisense transcripts was raised. Was this phenomenon related to Rhino and H3K9me3 accumulation observed in Figure 2DE?

The two studies were addressing different questions:

The Figure 2D-E was studying the molecular changes of a stable epigenetic conversion: that is to say the *BX2* locus was activated by heat treatment after several generations at 29°C and this activation is stable because it is maintained even when flies are incubated at 25°C (the classical temperature). In this context, we compared a stable OFF line and stable ON lines (converted by maternally inherited piRNAs or by temperature) for differences of chromatin marks. Statistical but weak differences were identified (Figure 2).

In the Figure 4, we were addressing the mechanism of the triggering, that is to say what is happening after one generation at 29°C. Therefore, in this experiment, we were not expecting to be able to identify chromatin differences with a 2% conversion rate.

Experiments in Figure 2 and Figure 4 should be performed using the same fly lines.

In Figure 2D-E, ChIP experiments were performed on flies stably activated by temperature. They hold the *P(TARGET)* since it is required for their activation and used as internal reporter of the activated state. In Figure 4, measurement of antisense *lacZ* transcripts from *BX2* implies that all *lacZ* RNA should come from *BX2* locus. The presence of *P(TARGET)* which shares *lacZ* sequence would impair the specific measurement of *lacZ* antisense coming from *BX2* locus.

4) Is the BX2 transgene in the BX2^OFF^ line containing P(TARGET) also inserted in the AGO1 gene? This should be examined. If this were the case, the authors better examine whether piRNAs were derived from the AGO1 gene.

All *BX2* transgenes, whether ON or OFF, are inserted into the *AGO1* gene. We have analyzed the small RNA produced from *AGO1* gene region in the *BX2^OFF^* and *BX2^ON^* strains: no significant piRNA amount was produced in all epigenetic contexts. This shows that *AGO1* is not a natural piRNA producing locus. Later, looking for piRNAs from *AGO1* gene region at 25°C and 29°C in *BX2^OFF^* strains confirm that the *AGO1* gene is not a piRNA cluster and that 29°C does not induce de novo production of small RNAs from the *AGO1* gene region. These data have been added in a new Figure 2—figure supplement 2 and in Figure 4J and are discussed in the text. Figure legends have been modified as well.

5) The authors should test experimentally whether BX2-derived piRNAs are loaded into PIWI proteins. This is very important.

From the first description of piRNAs and piRNA clusters by Brennecke et al., 2007, it is known that piRNAs are loaded by the PIWI proteins allowing their biogenesis. Through these loadings, piRNAs have four main signatures that defined piRNAs: 23-29 nt, a 1U bias, a 10A bias and a 10 nt overlap between pairs of 23-29 nt small RNAs (ping-pong signature). Since this first description, several other studies used these properties in numerous organisms (*Nematostella, Bombyx mori, D. simulans*, marmoset, ….) and identified piRNAs. The data of Figure 2 shows these characteristics that convince us that *bona fide* piRNAs were produced by the new activated *BX2* locus. On top of that, the strength of the transgene system used in this study allows also to provide the ultimate proof: the converted *BX2^theta^* locus is producing small RNA that are functionally active for germline *lacZ* repression.

6) Figure 2B: Explain why BX2-derived piRNAs did not show 10A bias. The data shown in Figure 2C supported the idea that BX2-derived piRNAs were products of the ping-pong pathway. Then, they should have shown 10A bias, but in reality they did not.

The 10A bias is only observed in the fraction of AGO3-loaded piRNA (Brennecke et al., 2007) or can be detected in the fraction of piRNA that paired each other (De Vanssay et al., 2012). We have performed this last analysis and the results are now shown on Figure 2C and Figure 2—figure supplement 1C. The figure legends and the Materials and methods section have been completed as well.

7) Figure 2DE: The authors should examine genome-widely where in the genome H3K9me3 and Rhino accumulated upon heat treatment. It is hard to imagine that H3K9me3 and Rhino accumulation only occurred at the BX2 locus upon heat treatment.

We agree that studying whole genome effects of heat treatment could be an interesting question. However, we did not detect production of new piRNA from other loci (Figure 2—figure supplement 3). Therefore, even if modification of chromatin marks were detected elsewhere in the genome, they are not followed by a stable new piRNA production, that is the question addressed in this study. Once again, molecular analyses were performed on flies that have been activated at 29°C and then put back at 25°C. Only stable modifications can be observed. Our results showed that only *BX2* is metastable for piRNA production, all other potential piRNA producing loci were already activated before heat treatment, and even if others loci are stably modified in their chromatin marks (H3K9me3 or Rhino), they did not become piRNA producing loci.

8) The authors should examine whether H3K9me3 and Rhino accumulated at the BX2 locus in BX2^OFF^ line without P(TARGET) upon heat treatment.

The low occurrence of epigenetic conversion (2%) does not allow enough dynamics to analyze chromatin modification. Without *P(TARGET)*, no conversion is observed and ChIP experiment on *BX2^OFF^* has already been performed and show less H3K9me3 or Rhino enrichment compared to *BX2** or *BX2^theta^*.

9) This manuscript should be edited thoroughly by a native English speaker.

The manuscript has been read and corrected by a native speaker (Dr Lori Pile).

10) Table 1 should be removed from the main text.

It has been done. Table 1 has been renamed Supplementary file 8

Reviewer #3:[…] While I have no problem with the experiments, but did wish for more insight into the mechanism, and more caution in the interpretation.

As previously described, we believe that the experiment with additional transgenes and with *Dcr-2* provide more insight related to the mechanistic part of the phenomenon that is illustrated by a model presented in Figure 5.

In particular, entirely enviromental specification of piRNA clusters implied in the Abstract is a strong claim, and as the authors show themselves, is an oversimplification. The induction they find is quite specific to their system. Specifically, there are higher levels of homologous sense and anti-sense transcript at 29 than at 25, and temperature alone doesn't change other piRNA production.

We agree that we oversimplified the message in the Abstract and as the first reviewer suggested, we now precisely state the role of different parameters required for the activation of *BX2*.

Similarly, it would be useful to know if the expression of the sense transcript from P-Target is also higher at 29 that at 25. (I think this construct has a heat-shock promoter, as well as that of the P-element, which shows temperature sensitive effects.)

We performed this experiment and the result is that *P(TARGET)* is less expressed at 29°C. This result has been added in the text and in a new supplementary figure (Figure 4—figure supplement 1C). As mentioned in the Materials and Methods, this transgene does not carry a heat shock promoter, it is an enhancer trap transgene inserted in the *eEF5* gene, an ubiquitous translation elongation factor.

I wasn't entirely convinced by the suggestion that expression of natural piRNA clusters is higher at 29C. Many piRNAs are generated as a secondary byproduct of transposable element message, and many transposable elements seem to have temperature sensitive expression (possibly as a byproduct of changes in chromatin.)

We do not suggest that expression of natural piRNA clusters is higher at 29°C. Actually, the analysis of the major *42AB* piRNA cluster shows a slight decrease of piRNA at 29°C compare to 25°C (Figure 4E). We totally agree that variations in piRNA production from natural piRNA clusters may result from several parameters including ping-pong amplification and variation in the expression of piRNA genes (as suggested by Fast et al., 2017). We have rephrased this part to clarify this point.

The piRNA dependent splicing suppression of the P-element is also, apparently, not temperature sensitive (Teixiera et al., 2017).

We note that the intron 2-3 of the *P* element, whose splicing is dependent of piRNA, is absent from all transgenes that have been used in this study: the *P(lacW)* transgene that constitutes the *BX2* locus, *P(TARGET)* and *P(A92)*, the two *P-lacZ-rosy* used as reporter transgenes.

Minor points. It does seem important (and quick) to eliminate the possibility that there's any active P-elements in these lines. While w1118 is a classic M-type background, there are some cases of sublines of other M-types harboring P-elements via contamination (Rahman et al. Nucleic Acids Research, Volume 43, Issue 22, 15 December 2015, Pages 10655-10672). If there is active P-element, I think it is possible that transgenes (which has the TIRs necessary for P-element insertion) has been picked up and put in another piRNA cluster. It's a simple matter of a few PCRs to exclude this possibility. The Cy and CyRoi backgrounds should be tested as well, to show that they are devoid of any plus-strand transcript that could be an ongoing trigger for piRNA production.

Several years ago we tested by PCR the absence of *P* element in the strains we used for our studies, especially in the balanced stocks commonly used. We checked again by PCR (using the primer 5'-TGATGAAATAACATAAGGTGGTCCCGTCG-3' known as P3-31 that recognized the *P* inverted repeated sequences) all of the stocks that we used in this study and all of the strains are devoid of natural *P* sequences. We propose to not add this result in the paper since it is a very specialized point but our results are available if needed. Another point to this remark is that we have established recombined lines from *BX2, P(TARGET)* and, in the absence of *P(TARGET)*, they did not show activation of the *BX2* locus at 29°C. If uncontrolled, natural *P* elements were present in our strain and responsible of the conversion, they should not all be eliminated by recombination and control lines could still show *BX2* activation. Additionally, we have published that piRNA produced by *P* sequences only are not able to activate *BX2* by paramutation (Hermant et al., 2015) demonstrating that the homology length conferred by *P* sequences is not sufficient to trigger *BX2* conversion, even by piRNA maternal inheritance.

Subsection “Epigenetic conversion at 29°C occurs at a low rate from the first generation” paragraph two. The model is entirely sensible, but the data are quite noisy, and seem like they would be consistent with any model predicting an increase over generations. Can they compare this model to, for example, a model without the “c” parameter using Akaike Information Criteria or a log-likelihood test? This would make the analysis more informative.

We agree that the data are noisy. This is due to the high variation of occurrence of the epigenetic conversion and its sampling through generations and among replicates. The first rough observations of the phenomenon reported complete repression in ovaries after several generations. We tried to understand this effect by analyzing thousands of flies in one generation rather that dozens of flies through multiple generations and observed a small conversion frequency (≈2%, Figure 3). Then, we searched for a simple model based on the conversion frequency we observed in one generation and the fact that the ON state is transmitted to the next generation (De Vanssay et al., 2012 and Figure 3). The major problem we encountered is related to the sampling that occurred at each generation: among all egg chambers that are produced, only dozens of flies were analyzed. We think that the sampling size generates the variation we observed among replicates with a 2% conversion phenomenon. We agree with the reviewer that our data would fit with a number of models describing an increase over generations. We can remove this part if it is still considered non informative.

Subsection “Heat conversion requires a homologous sequence in trans”: The chi-square statistic and degrees of freedom should be reported as well as the p-value. (And this is very minor, but the Greek letter is usually used in place of “chi”.)

We have added in the new version the χ^2^ value (23.35) and the degrees of freedom (2) and replaced “chi2” by “χ^2^” each time we used the test.

[Editors' note: further revisions were suggested, as described below.]

The reviews follow below, and we will look forward to hearing how you propose to proceed.

Reviewer #1:

In the revised version of the manuscript the authors find an involvement of sense-antisense transcripts and Dicer-2 in piRNA biogenesis. The authors may remember that one of the early findings celebrated as a major discovery was the lack of a role for double-stranded RNAs and the double-stranded RNA cleaving enzyme Dicer, in piRNA biogenesis. If now, the authors find a role for these in piRNA biogenesis, it has to be backed with strong data.1) Can the authors demonstrate loading of the 21 nt RNAs into Piwi proteins?2) If its role is epigenetic conversion, then ideally it has be loaded into the nuclear Piwi.3) Are there experiments linking Dicer-2 to loading into a Piwi protein? Are they ever found in the same complexes?I agree that there is a strong functional/genetic data that is mysterious, but very interesting. The heat-activated piRNAs are functional as they can silence a LacZ transgene. Since this manuscript will be published in some form, I am happy to support description of this part in a revised manuscript, but without wild speculation of Dicer-2 in piRNA biogenesis (even if they examine a Dicer mutant). These attempts at explaining the molecular mechanism may be toned down. I am worried that any prominent mention of Dicer-2 in this context may only serve to foster suspicion of the interesting genetic observation.

We agree with the first reviewer that implications of *Dcr-2* in piRNA biology suggest a major breakthrough that our experiments do not definitively prove. We would like to clarify, however, that the role of *Dcr-2* we proposed was not in the maintenance of piRNA production, but rather, a more opportunistic way to accidentally produce some new piRNAs. In our system, where the *BX2* cluster is a peculiar genomic structure, this de novo piRNAs production would be sufficient to convert *BX2* into an active piRNA cluster. Once activated, *Dcr-2* is no longer required for piRNA production from *BX2* as we published in Hermant et al., 2015. We agree that the biochemical experiments proposed by the first reviewer are well adapted in theory. We think, however, that the 2% conversion rate is too low to be able to observe such molecular details. For these reasons, we agree to your proposal to remove the results obtained with *Dcr-2* and tone down our model presented in Figure 5. We propose to consider the hypothesis of somehow making double strand RNAs as a starting point to accidentally produce the first new piRNAs, but clearly state that much of the mechanistic aspects remain unknown, as asked by the editors.

Reviewer #3:

I still find this study clever and interesting, and think the manuscript shows some specific improvements:The “primed” nature of the BX2 locus is clearer, and the claim in the paper is more measured and reasonably justified.The experiment showing the Dcr2 dependence of the conversion of the BX2 from off to on goes a little way toward understanding the mechanism of piRNA cluster formation, implying that it is dependent on siRNAs.The addition of the figure showing the model for the mechanism. The weak point of the proposed mechanism, as I see it, is the lack of evidence that the siRNAs are loaded onto PIWI, any evidence the authors can provide for this mechanism would greatly improve the paper. I can't make any suggestions that seem technically feasible, however, in light of the low conversion rate in this system.

As stated above, we agree that the biochemical experiments proposed by the reviewer are well adapted in theory but that the 2% conversion rate is too low to be able to observe such molecular details. For these reasons, we agree to your proposal to remove the results obtained with *Dcr-2* and tone down our model presented in Figure 5.

Further clarification would also be helpful for these aspects:Regarding the model in Figure 1—figure supplement 3: I previously suggested a statistical analysis of this model. Rather than the more detailed analysis I suggested previously, I think a reasonable compromise would be to show that there is a significant increase over generations. That is the important point here (and, in fact, it's hard to imagine a scenario where the “memory” of conversion does not play a role in the increase over generations). But, while there appears to be a trend, the data are sufficiently noisy that it would be useful just to see that the increase is significant (perhaps a linear regression on transformed data?).

We removed the mathematical model of progression of conversion during generation and propose a simpler description of the variability observed among lines. Mean and 95% confidence interval per generation were calculated on modified data (arcsine square root). The text (subsection “Epigenetic conversion at 29°C occurs at a low rate from the first generation”), the Figure 1—figure supplement 3B and its legend have been modified accordingly.

Discussion opening paragraph: Please explain what you mean by the “specific spliced transcript” comment; it wasn't clear.

By “specific spliced transcript”, we mean that we measured *AGO1* transcripts that overlap the *BX2* insertion point and that come from a distant promoter. We have added this specific description in the legend of Figure 4. There is another promoter for *AGO1* that is located after the *BX2* insertion point. *AGO1* transcripts produced from this promoter are only slightly affected by temperature (not significantly), allowing us to conclude that it could explain the apparent discrepancy with the results published in Fast et al., 2017. We have added this analysis in Figure 4—figure supplement 1D and modified the discussion and the legend, accordingly.

e.g., supplementary file 11: It would be nice to more cautious in the interpretation where there are large differences in the number of flies examined. In Figure 3, for example, I think the numbers show that 41 of 1447 females show partial or complete repression; the comparable (?) numbers is Supplementary file 11 show 0 of 32 (?) females show any repression. This is not a significant difference via Fisher's exact test.

We would like to clarify that the results obtained with NaCl and heat shock have to be compared to those obtained at 29°C in G1 (showed in Figure 3), considering the proportion of converted egg chambers and not flies (that are analyzed in G2). For the heat shock experiment, the Χ^2^ test was 106.03 and the p-value was 7.25 x 10^-25^ (heat shock 0/3840 versus 29°C 586/21720) and for the NaCl experiment, the Χ^2^ test was 115.93 with a p-value of 4.91 x 10^-27^ (NaCl 0/4200 versus 29°C 586/21720). We have described this analysis in a clearer way in the legend of supplementary file 11 and by adding the results obtained at 29°C (from Figure 3) in the table of the revised version.

Finally, we renamed the *P(TARGET)* transgenes used in this study as asked by editors and specified their domain of expression to be as clear as possible for a large audience.